# Catecholaminergic modulation of meta-learning

**Jennifer L Cook[1]\*, Jennifer C Swart[2], Monja I Froböse[2], Andreea O Diaconescu[3,4,5], Dirk EM Geurts[2,6], Hanneke EM den Ouden[2†], Roshan Cools[2,6†]**

[1]School of Psychology, University of Birmingham, Birmingham, United Kingdom; [2]Donders Institute for Brain, Cognition and Behaviour, Centre for Cognitive Neuroimaging, Radboud University, Nijmegen, Netherlands; [3]Translational Neuromodeling Unit, Institute for Biomedical Engineering, University of Zurich and ETH Zurich, Zurich, Switzerland; [4]Department of Psychiatry, University of Basel, Basel, Switzerland; [5]Krembil Centre for Neuroinformatics,CAMH, University of Toronto, Toronto, Canada; [6]Department of Psychiatry, Radboud University Medical Centre, Nijmegen, Netherlands

**\*For correspondence:**
j.l.cook@bham.ac.uk

[†]These authors contributed equally to this work

**Abstract** The remarkable expedience of human learning is thought to be underpinned by meta-learning, whereby slow accumulative learning processes are rapidly adjusted to the current learning environment. To date, the neurobiological implementation of meta-learning remains unclear. A burgeoning literature argues for an important role for the catecholamines dopamine and noradrenaline in meta-learning. Here, we tested the hypothesis that enhancing catecholamine function modulates the ability to optimise a meta-learning parameter (learning rate) as a function of environmental volatility. 102 participants completed a task which required learning in stable phases, where the probability of reinforcement was constant, and volatile phases, where probabilities changed every 10–30 trials. The catecholamine transporter blocker methylphenidate enhanced participants' ability to adapt learning rate: Under methylphenidate, compared with placebo, participants exhibited higher learning rates in volatile relative to stable phases. Furthermore, this effect was significant only with respect to direct learning based on the participants' own experience, there was no significant effect on inferred-value learning where stimulus values had to be inferred. These data demonstrate a causal link between catecholaminergic modulation and the adjustment of the meta-learning parameter learning rate.

## Introduction

The remarkable expedience of human learning has given rise to the theory of meta-learning, or 'learning to learn' (*Botvinick et al., 2019*; *Duan et al., 2016*; *Wang et al., 2016*), which postulates hierarchically nested learning systems wherein 'slow' trial-and-error-based learning systems consolidate abstract rules, while meta-learning mechanisms rapidly adjust to the current learning environment. This combination enables humans to learn generalities that commonly occur across spatial and historical contexts whilst also being able to rapidly optimise learning within a specific context. To date, the human neurobiological implementation of meta-learning remains unclear. A burgeoning literature argues that the catecholamines dopamine and noradrenaline play an important modulatory role in meta-learning, and thus predicts that altering catecholamine function should change the ability to adjust learning to suit the prevailing context. The current study tests this hypothesis.

Relatively simple learning algorithms may be sufficient to predict choice behaviour in a static environment. In real life, however, learning environments are almost always dynamic. Thus, learners must

'meta-learn', that is they must adjust their learning strategies to suit the current environment (*Lee et al., 2004*; *Lee et al., 2014*; *Soltani et al., 2006*). A classic example of human meta-learning concerns the adjustment of learning rate as a function of environmental volatility. In environments where payoff probability fluctuates (volatile environments), learning rate should be high, favouring recent over (outdated) historical outcomes. However, in stable environments, a low learning rate is optimal since recent and historical outcomes provide similarly useful information and a low learning rate prevents spurious, contradictory, outcomes having a disproportionate impact. Indeed, numerous studies have demonstrated that human participants adjust learning strategies depending on the volatility of their environment (*Behrens et al., 2007*; *Behrens et al., 2008*; *Lee et al., 2014*; *McGuire et al., 2014*), typically adopting higher learning rates in volatile relative to stable environments (*Browning et al., 2015*; *Krugel et al., 2009*; *Nassar et al., 2010*). Thus, humans adjust learning rates to suit the current level of environmental volatility.

Although the neural mechanisms underpinning this example of meta-learning are unclear, various neuro-computational models have been proposed. These models differ in terms of their precise computational mechanisms. However, they converge on the hypothesis that catecholamines play a central role in human meta-reinforcement learning. One example concerns a class of models that propose that the prefrontal cortex comprises a meta-learning system that, via striatal dopaminergic prediction error signals (*Montague et al., 1996*; *Schultz et al., 1997*), learns task structure and extracts abstract rules (*Badre and Frank, 2012*; *Collins and Frank, 2013*; *Collins and Frank, 2016*; *Frank and Badre, 2012*; *Lieder et al., 2018*; *Massi et al., 2018*; *Pasupathy and Miller, 2005*; *Wang et al., 2018*). This class of models proposes a critical role for dopamine in training the dynamics of the prefrontal meta-learning system. A parallel theory developed by Doya and colleagues also highlights the importance of dopamine: Doya and colleagues propose that learning rate should be selected based on a formal comparison between medium- and long-term running averages of reward rate (*Doya, 2002*; *Schweighofer and Doya, 2003*), which are hypothesised to be carried in the phasic and tonic firing of dopamine neurons, respectively (*Schweighofer and Doya, 2003*). Finally, *Silvetti et al. (2018)* emphasise the importance of noradrenaline, suggesting that the locus coeruleus (the main noradrenergic nucleus of the brain) is the primary driver in computing learning rates. In sum, the neuro-computational modelling literature proposes that the catecholamines are central to human meta-reinforcement learning.

In keeping with theory, preliminary empirical evidence from the human psychopharmacological literature is beginning to confirm the suggested role for the catecholamines in the adjustment of learning rates. For example, a number of studies have shown that learning rates increase with increasing catecholamine availability (*Howlett et al., 2017*; *Jepma et al., 2016*; *Marshall et al., 2016*), and decrease with reduced catecholamine availability (*Diederen et al., 2017*). However, meta-learning theories (*Collins and Frank, 2016*; *Schweighofer and Doya, 2003*; *Silvetti et al., 2018*; *Wang et al., 2018*) predict that enhancing catecholamine function should not simply increase (or decrease) learning rate but rather modulate the ability to adjust the learning rate to suit the current learning environment, potentially via effects on the ability to learn the abstract rules governing the structure of the environment (*Collins and Frank, 2016*; *Wang et al., 2018*). Indeed, *Wang et al. (2018)* have demonstrated that, when trained on a two-armed bandit task featuring volatile and stable phases (*Behrens et al., 2007*), their computational model can rapidly infer the volatility of the current environment and adjust learning rate accordingly. Existing studies of catecholaminergic effects on learning rate adjustments have either explicitly signalled changes in the learning environment (*Diederen et al., 2017*), and therefore removed any need to learn the task structure, or have used volatile learning environments where constant high, but not low, learning rates are adaptive, that is where no learning rate adjustments are required (*Howlett et al., 2017*; *Jepma et al., 2016*; *Marshall et al., 2016*). In contrast, here we use a task with changing levels of volatility (e.g. *Behrens et al., 2007*), such that low learning rates are optimal in stable phases and higher learning rates are optimal in volatile phases. This allowed us to test the prediction that enhancing catecholamine function modulates the ability to adjust learning rate as a function of environmental volatility.

The abovementioned studies implicate catecholamines in the adjustment of learning rate with respect to direct 'experienced-value' learning, where choices are based only on participants' own previous experience with outcomes given their actions. Experience-value learning can be contrasted with indirect 'inferred-value' learning, in which stimulus value must be inferred from integration of one's own experience of outcomes with knowledge of the environment. For example, advice from a

colleague may not have inherent value (i.e. may not be directly rewarding/punishing), but the colleague's advice may promote an action which is directly rewarded. Using the knowledge that the colleague's advice promoted the rewarded action, one can infer the utility of the advice and update its corresponding value estimate. Existing studies implicate the catecholamines in this process (*Langdon et al., 2018*; *Sharpe and Schoenbaum, 2018*). For instance, in reversal tasks where a change in a directly experienced stimulus value indicates that the value of an unchosen option has also changed, dopamine neurons in the monkey brain signal both the experienced-value, and the inferred-value of the unchosen option (counterfactual updating; *Bromberg-Martin et al., 2010*). Recent meta-learning simulations also produce this 'inferred-value effect' (*Wang et al., 2018*). Despite this nascent literature, human psychopharmacological studies have not empirically investigated catecholaminergic modulation of learning rates with respect to inferred-value learning, but in theory it should be the case that catecholamines are also involved in the modulation of learning rate with respect to inferred-value.

Participants in our study completed a modified version of the two-armed bandit task developed by *Behrens et al. (2008)* (see also *Cook et al., 2014*). On each trial, participants chose between two bandits which fluctuated between stable phases in which the probability of reinforcement was held constant for 50–60 trials, and volatile phases in which probabilities changed every 10–30 trials. Preceding the response phase, a frame surrounded one of the two bandits, 'advising' participants to select the highlighted bandit. The veracity of the frame's 'advice' similarly fluctuated between stable and volatile phases which were independent of the fluctuating volatility of the bandits. Experience with outcome information **directly** informed participants about the reward value of each bandit and **indirectly** informed participants about the value of the advice. That is, we instructed participants that the frame represents 'advice' and that this 'advice' can be either correct or incorrect. Informed by a parallel literature concerning differences between social and non-social learning (*Heyes, 2012*), half of the participants were informed that the frame represented social information, while the other half received a non-social cover story. This latter manipulation enabled us to test for any social-specificity regarding the source of the 'advice'. To investigate the effect of a catecholamine challenge on learning rate adjustment for experienced- and inferred-value learning, participants completed the task once under placebo (PLA) and once after 20 mg of the catecholamine transporter blocker methylphenidate (MPH; order counterbalanced). Whilst potential differences between social and non-social advice were our initial concern (which we discussed in depth in *Cook et al., 2018*), in response to major recent advances in the meta-learning literature, we here focus on the effects of the catecholaminergic challenge on meta-learning and briefly discuss comparisons between social and non-social groups for completeness.

To estimate learning rate adjustment as a function of environmental change, we fitted an adapted Rescorla-Wagner learning model (*Rescorla and Wagner, 1972*). Our adapted model estimates learning rate for directly-experienced- and indirectly-inferred-value learning, for volatile and stable learning environments. To foreshadow our results, we observed that MPH improved participants' ability to adjust learning rate as a function of environmental volatility. However, this effect was significant only with respect to experienced-value learning: MPH had no significant effect on (either social or non-social) inferred-value learning. This pattern of results was also reflected in increased use of a win-stay, lose-shift strategy in volatile relative to stable environments, under MPH, relative to PLA. Again, we observed that MPH only significantly affected the use of a win-stay lose-shift strategy for experienced-value learning. In sum, these results demonstrate the causal role of catecholamines in meta-reinforcement learning as evidenced by enhancement of the adjustment of learning rate to suit the level of environmental volatility.

## Results

102 participants took part on two separate test days. The study followed a double-blind cross-over design such that 50% of participants received 20 mg MPH on day 1 and PLA on day 2, for 50% of participants the order was reversed. On each test day, participants completed the two-armed bandit task (*Figure 1*) (*Behrens et al., 2008*; *Cook et al., 2014*). On each trial participants chose between a blue and a green stimulus to accumulate points. The probability of reward associated with the blue (p(blue)) and green stimuli (p(green)) varied according to a probabilistic schedule (max. 0.8, min 0.2). P(green) was always equal to 1 - p(blue). Participants were randomly allocated to one of four

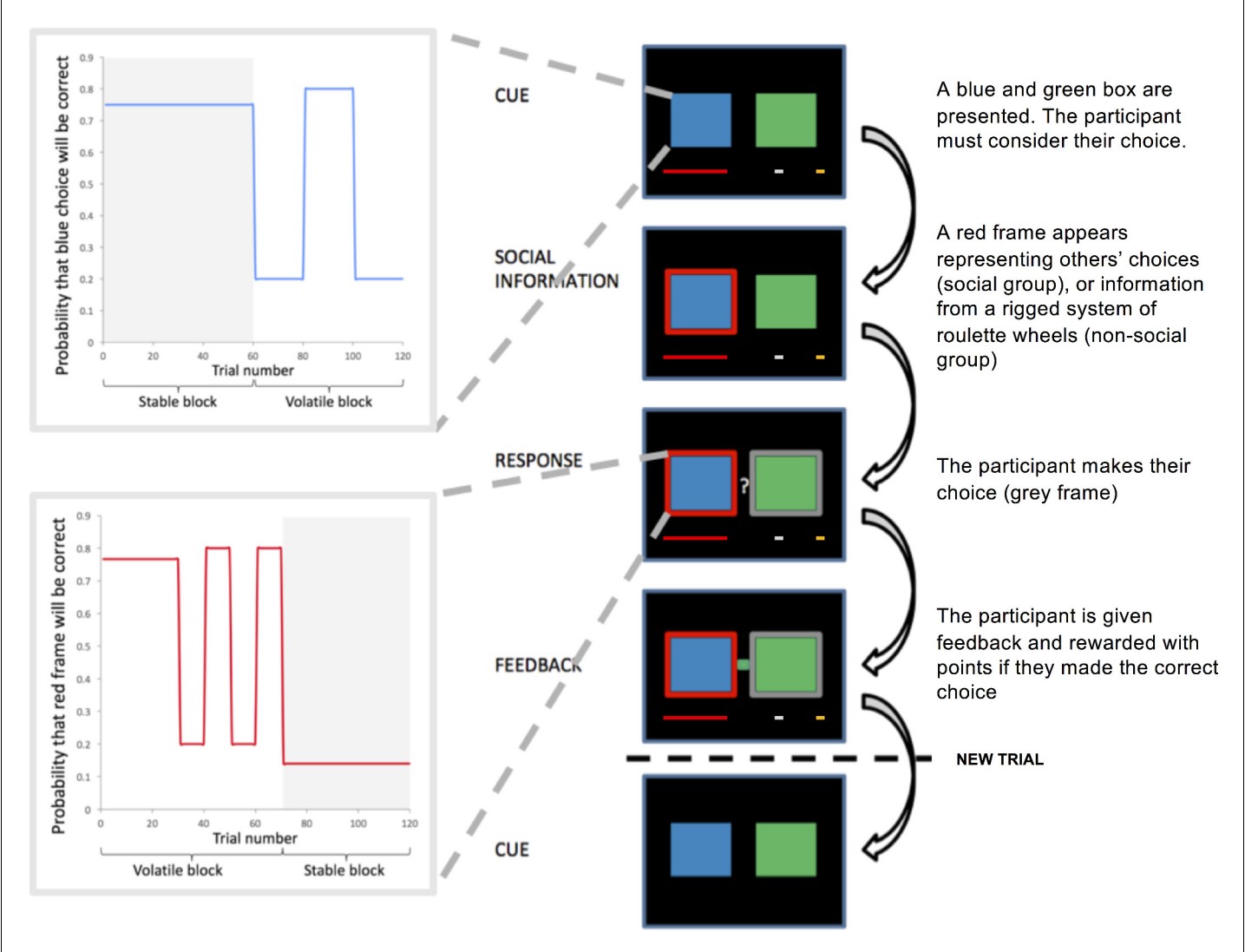

**Figure 1.** Task design. Participants selected between two bandits (blue and green boxes) in order to win points. On each trial, participants saw the direct sources (boxes, 1–4 s), subsequently either the blue or green box was highlighted with a red frame (the indirect source, 1–4 s). Participants were instructed that the frame represented either the most popular choice made by a group of participants who had completed the task previously (Social Group) or the choice from rigged roulette wheels (Non-social Group). After participants had responded, their selected option was framed in grey. After 0.5–2 s participants received feedback in the form of a green or blue box between the two bandits. A new trial began after 1–3 s. Success resulted in the red reward bar progressing towards the silver and gold goals. The probability of reward associated with the blue and green boxes and the probability that the red frame surrounded the correct box varied according to probabilistic schedules which comprised stable and volatile phases. The online version of this article includes the following figure supplement(s) for figure 1:

**Figure supplement 1.** Probabilistic schedules.

possible probabilistic schedules (see Materials and methods: Learning task). All probabilistic schedules were comprised of stable phases, in which the probability of reinforcement was held constant for 50–60 trials, and volatile phases, in which probabilities reversed every 10–30 trials. Outcomes were pre-determined according to these schedules. Although the probabilistic schedule was the same across the two study days, there was variation in the trial-wise outcomes. Participants could use their direct experience of the reward history to track these changing probabilities (experienced-value learning). On each trial, before participants made a response, they saw an 'advice' frame surrounding one of the two options. The veracity of this information followed one of four different probabilistic schedules (*Figure 1—figure supplement 1*) that participants could track throughout the course of the experiment. As with the blue and green stimuli, the probabilistic schedule

underpinning the veracity of the frame comprised stable phases in which the probability that the frame surrounded the correct option was held constant for 50–60 trials, and volatile phases in which probabilities changed every 10–30 trials. Participants were randomly allocated to one of two experimental groups, 'Social' or 'Non-social'. There were no significant differences between groups in age, gender or socioeconomic status (Appendix 1). Participants in the Social Group were informed that the frame represented the most popular choice selected by a group of participants who had previously played the task. They were also told that the experimenter had 'mixed up' the order of the other participants' trials so that their advice follows useful and less useful phases. Participants in the Non-social Group were instructed that the red frame represented the outcome from a system of rigged virtual roulette wheels, which fluctuated between providing useful and less useful advice (see Appendix 2 for instruction scripts). Outcome information was represented by a blue or green indicator which directly informed participants about whether they had made the correct choice on the current trial (i.e. if the outcome indicator was blue, then the blue stimulus was correct) and from which they could indirectly infer the veracity of the frame: If the outcome was blue AND the frame surrounded the blue stimulus, then the advice had been correct. Unbeknownst to participants, reward was probabilistic, so that in a stable blue phase (e.g. p(blue) = 0.75), green responses were occasionally rewarded (e.g. on 25% of trials).

## Accuracy analysis

To account for the probabilistic nature of feedback, correct responses were defined as those choices that were likely to be rewarded according to the probabilistic schedule and incorrect responses as those that were unlikely to be rewarded. Overall accuracy was mean (standard error) ($\bar{x}(\sigma_{\bar{x}})$) = 0.69 (0.01; min. 0.44, max. 0.89) under PLA and 0.69 (0.01; min. 0.52, max. 0.87) under MPH. A repeated measures ANOVA with within-subjects factors drug (MPH, PLA) and volatility (stable, volatile), and between-subjects factor group (social, non-social) revealed a significant main effect of volatility (F (1,100) = 20.904, p < 0.001), with accuracy being significantly higher for stable ($\bar{x}(\sigma_{\bar{x}}) = 0.711$ (0.011)) relative to volatile phases ($\bar{x}(\sigma_{\bar{x}}) = 0.664(0.009)$), and no other main effects or interactions (all p>0.05).

## Win-stay, lose-shift analysis

A simple learning strategy is the win-stay and lose-shift (WSLS) strategy. A participant who unerringly follows a WSLS strategy is completely guided by their performance on the previous trial: If they choose blue and win they *stay* with a blue choice; if they choose blue and lose, they *shift* to a green choice. In a classic reinforcement learning model (e.g. *Rescorla and Wagner, 1972*) such a participant would exhibit a learning rate equal to 1. That is, they would be completely 'forgetting' about past events and reacting only to trial t-1. The extent to which participants employ a WSLS strategy typically correlates with their learning rate, thus WSLS can be (cautiously) used as a proxy for learning rate. Our first analysis therefore investigated the influence of catecholaminergic modulation of meta-learning by quantifying the effect of MPH on the use of this simple learning strategy in volatile and stable environments. In stable phases, performance depends on ignoring misleading probabilistic feedback, thus lose-shifting would result in worse performance as participants would stray from the optimal action every time they receive a spurious (probabilistic) punishment. In contrast, in a volatile phase a punishment is relatively more likely to signal a true change in the underlying probabilistic schedule and thus lose-shifting is adaptive and should be more likely than in a stable phase. We predicted that, relative to PLA, MPH should enhance the use of a WSLS strategy in volatile compared to stable environments.

WSLS behaviour was quantified using a regression on individual participant choices (green/blue; follow/deviate from advice), to assess the degree to which a WSLS strategy guided these choices. For each participant, a regressor was created representing the choice that they would have made, for each trial, if they were strictly adhering to a WSLS strategy, for each of the eight cells of the 2x2x2 factorial design (drug x volatility x learning type). The resulting estimated $\beta_{WSLS}$ values were analysed with a repeated measures ANOVA with within-subjects factors drug (MPH, PLA), volatility (stable, volatile), and learning type (experienced, inferred) and between-subjects factor group (social, non-social). We observed a significant drug x volatility x learning type interaction (F(1,100) = 8.171, p = 0.005; *Figure 2C*). Post-hoc ANOVAs conducted separately for experienced-value and

inferred-value learning (*Figure 2A, B*) revealed a significant drug x volatility interaction for experienced-value learning (F(1,100) = 5.130, p = 0.026), and no significant interaction (or main effect of drug) for inferred-value learning (F(1,100) = 2.310, p = 0.132; $BF_{01}$ = 3.5). Simple effects analyses revealed that, for experienced-value learning, $\beta_{WSLS}$ values were higher for volatile relative to stable phases under both PLA (volatile $\bar{x}(\sigma_{\bar{x}})$ = 0.408(0.023), stable $\bar{x}(\sigma_{\bar{x}})$ = 0.342(0.023), F(1,100) = 8.803, p

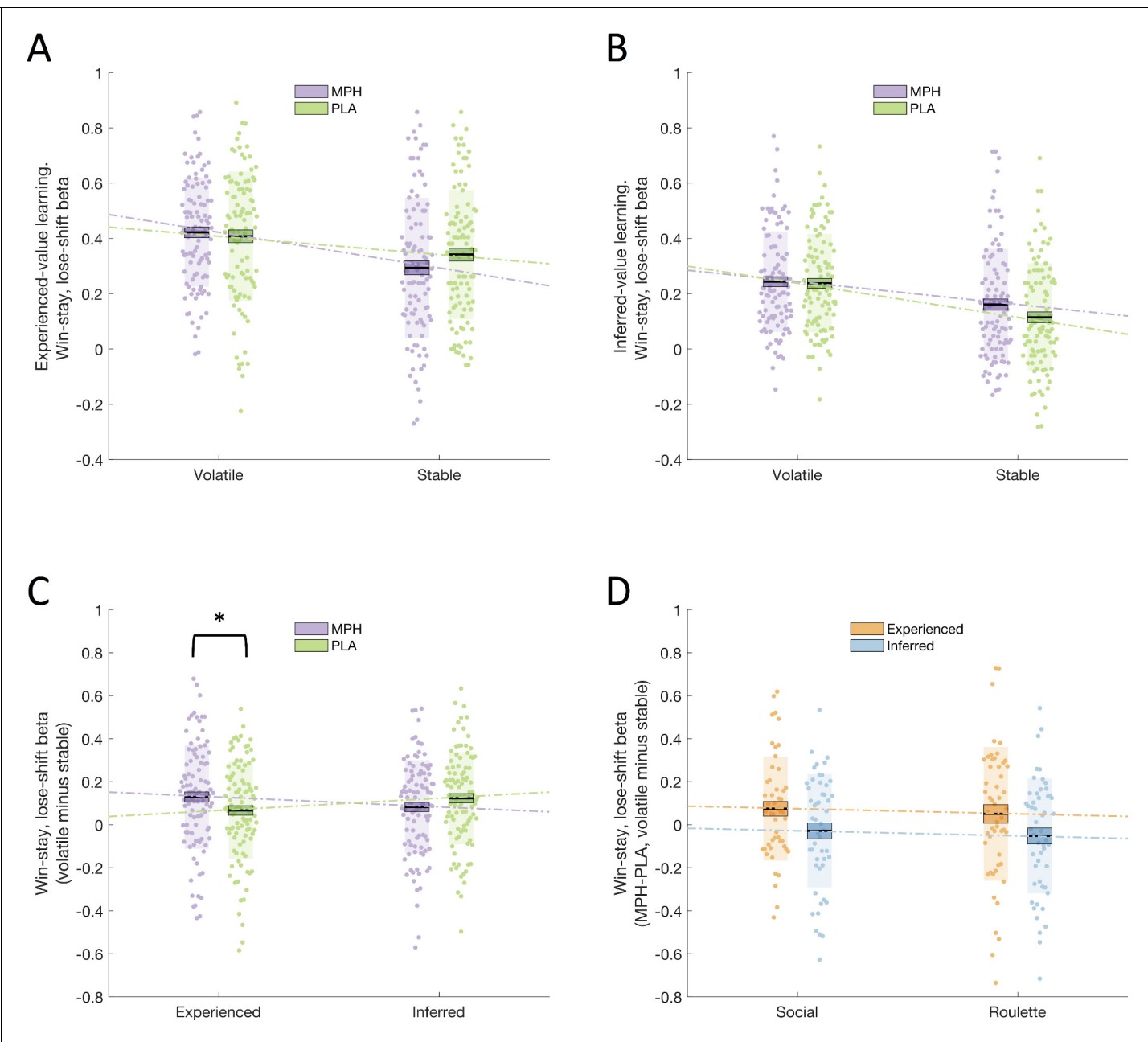

**Figure 2.** Drug effects on win-stay, lose-shift beta values. (**A**) Win-stay, lose-shift betas, for experienced-value learning, in stable and volatile periods, under MPH (purple) and PLA (green). (**B**) Win-stay, lose-shift betas, for inferred-value learning, in stable and volatile periods, under MPH and PLA. (**C**) There was a significant interaction between drug and volatility for experienced-value learning, but not for inferred-value learning. (**D**) Drug effect on win-stay, lose-shift betas for volatile minus stable blocks, for experienced-value (orange) and inferred-value (blue) learning. There was no difference between the social and non-social (roulette) groups. Boxes = standard error of the mean, shaded region = standard deviation, individual datapoints are displayed, MPH = methylphenidate, PLA = placebo, * indicates statistical significance.

The online version of this article includes the following figure supplement(s) for figure 2:

**Figure supplement 1.** Illustration of relationship between experienced-value $\beta_{WSLS\_vol-stable}$ and accuracy.

= 0.004) and MPH (volatile $\bar{x}(\sigma_{\bar{x}})$ = 0.422(0.020), stable $\bar{x}(\sigma_{\bar{x}})$ = 0.293(0.025), F(1,100) = 29.164, p < 0.001). Simple effects analyses also showed that, for experienced-value learning, there was a non-significant trend towards lower $\beta_{WSLS}$ values for MPH compared to PLA within stable (F(1,100) = 3.405, p < 0.068) but not volatile phases (F(1,100) = 0.427, p = 0.515). As demonstrated by the significant interaction, the difference between volatile and stable was greater under MPH ($\bar{x}(\sigma_{\bar{x}})$ = 0.129(0.024)) compared with PLA ($\bar{x}(\sigma_{\bar{x}})$ = 0.067(0.022).

There was no significant drug x volatility x learning type x group interaction (F(1,100) < 0.001, p = 0.994; *Figure 2D*) suggesting that the drug effect did not vary as a function of (social/non-social) group. In addition, there was no drug x group (F(1,100) = 0.095, p = 0.758) or drug x volatility x group (F(1,100) = 0.356, p = 0.552) interaction. To provide a test of whether it is accurate to claim that drug effects did not differ as a function of group, results were also analysed within a Bayesian framework using JASP (*JASP Team, 2018*) with a prior of a medium effect size using default priors. Bayes Factors provide a ratio of the likelihood of the observed data under the null versus alternative hypothesis (*Dienes, 2016*). Bayes Factors (BF$_{01}$) are reported. By convention, values of 0.33–0.1 and 3–10 are taken as moderate evidence in favour of the alternative and null hypotheses, respectively, 0.33–1 and 1–3 are taken as anecdotal evidence, while values of 1 are judged to provide no evidence to favour either the null or alternative hypothesis (*Lee and Wagenmakers, 2014*). We computed one single difference score representing the differential effects of MPH on experienced- and inferred-value learning in stable and volatile environments. This score was calculated separately for Social and Non-social Groups and submitted to a Bayesian independent samples t-test. The BF$_{01}$ value was 4.3 thus providing support for the null hypothesis that the groups do not differ.

There were no other significant interactions involving the factor drug and no main effect of drug (all p>0.05). See Appendix 3 for all other main effects and interactions and separate analyses of WS and LS scores. The interaction between drug, volatility and learning type remained significant (F(1,98) = 7.004, p = 0.009) when order of drug administration (MPH Day 1, PLA Day 1) was included as a factor in the ANOVA. There was no drug x volatility x learning type x order interaction (F(1,98) = 1.100, p = 0.297) and no main effect of order (F(1,98) = 0.218, p = 0.641).

## MPH significantly affects learning rate adaptation for experienced-, not inferred-, value learning

### Learning rate analysis

The above analysis demonstrates that, relative to PLA, MPH increased the use of a WSLS strategy in volatile versus stable phases. As noted above, a participant who unerringly follows a WSLS strategy is completely guided by their performance on the previous trial. In a classic reinforcement learning model (e.g. *Rescorla and Wagner, 1972*), this is equivalent to a learning rate equal to 1, where the participant would be completely 'forgetting' about past events beyond the preceding trial. In contrast, a lower learning rate indicates more equal weighting of recent and distant past. Consequently, the WSLS analysis (above) may indicate that MPH enhances the difference in learning rate between volatile and stable phases such that learning rates are higher in volatile compared with stable environments. To explore this possibility, we estimated learning rate by fitting an adapted Rescorla-Wagner learning model (*Rescorla and Wagner, 1972*). This learning model consisted of two Rescorla-Wagner predictors (*Rescorla and Wagner, 1972*), one that estimated the utility of a blue choice based on experienced-value learning in volatile and stable phases, and one that estimated the utility of going with the red frame (i.e. 'following the advice') using inferred-value learning in volatile and stable phases. These modelled choice values were translated into choices using a unit square sigmoid function, where the inverse temperature parameter $\beta$ controls the extent to which action value differences influence choice. This function also included a free parameter $\zeta$, which controls the weighting of experienced relative to inferred values (see Materials and methods for further details). Thus, per participant, we estimated four learning rates ($\alpha$), per drug condition: $\alpha_{stable\_experienced}$, $\alpha_{vol\_experienced}$, $\alpha_{stable\_inferred}$, $\alpha_{volatile\_inferred}$ plus $\beta$ and $\zeta$ (see Materials and methods for details regarding model fitting and model simulations. See Appendix 4 for results of model comparison). Paired samples t-tests showed no difference in $\beta$ and $\zeta$ estimates between MPH and PLA conditions: $\beta$ (MPH $\bar{x}(\sigma_{\bar{x}})$ = 3.354(0.209), PLA $\bar{x}(\sigma_{\bar{x}})$ = 3.088(0.167); t(101) = 1.152, p = 0.252; $\zeta$ (MPH $\bar{x}(\sigma_{\bar{x}})$ = 0.499(0.027), PLA $\bar{x}(\sigma_{\bar{x}})$ = 0.512(0.027); t(101) = -0.554, p = 0.581).

We tested the effect of MPH on learning rates using a repeated measures ANOVA with within-subjects factors drug (MPH, PLA), volatility (stable, volatile), and learning type (experienced, inferred) and between-subjects factor group (social, non-social). Supporting our WSLS analyses, we found a significant drug x volatility x learning type interaction (F(1,100) = 6.913, p = 0.010; *Figure 3C*). Again, post-hoc ANOVAs conducted separately for experienced-value and inferred-value learning (*Figure 3A, B*) revealed a significant drug x volatility interaction for experienced-value learning (F(1,100) = 6.151, p = 0.015), and no interaction (or main effect of drug) for inferred-value learning (F(1,100) = 1.588, p = 0.211, $BF_{01}$ = 3.3). Simple effects analyses showed that, for experienced-value learning, there was a non-significant trend towards lower $\alpha_{experienced}$ values for MPH

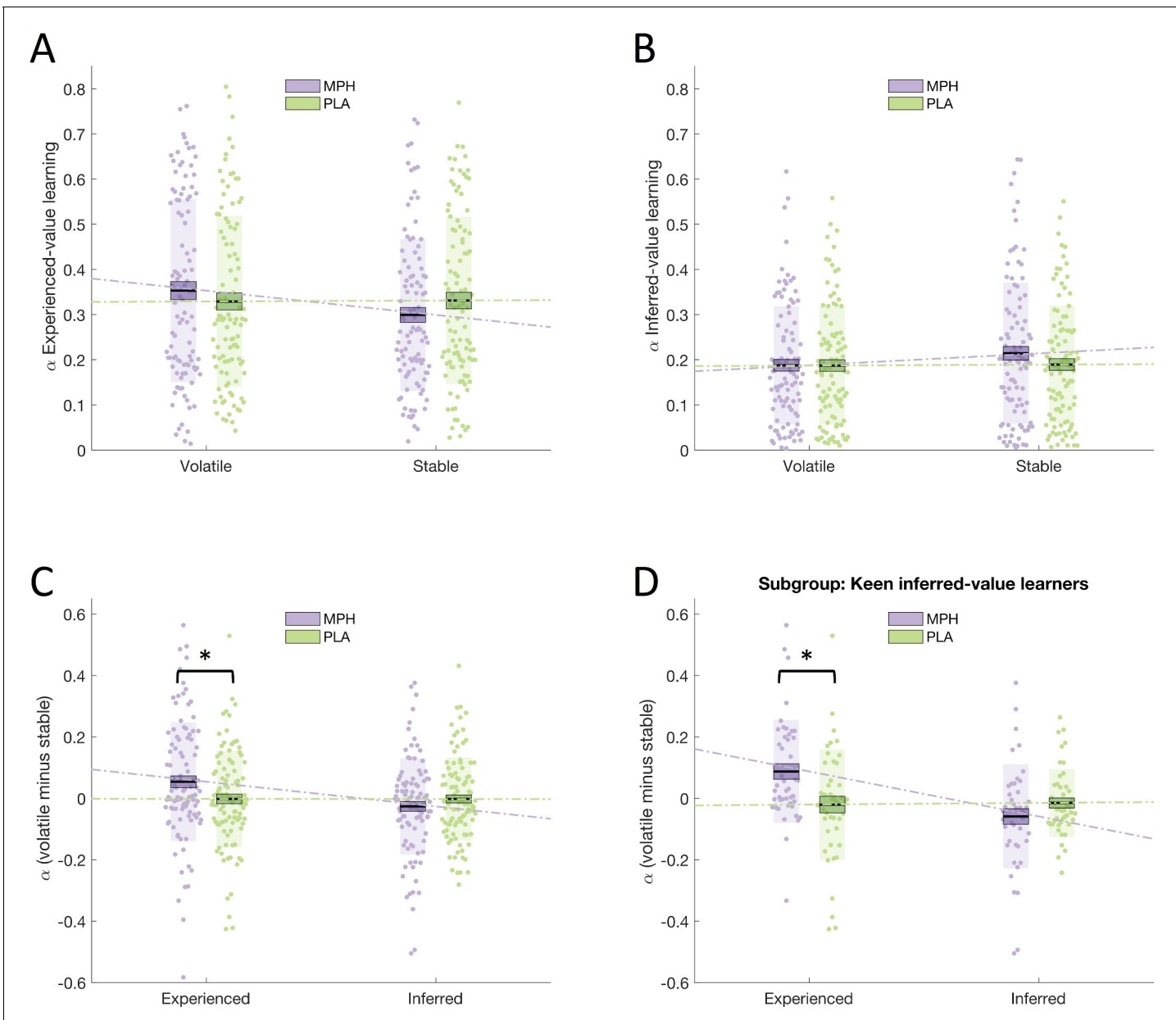

**Figure 3.** Drug effects on learning rates. (A) Learning rate for experienced-value learning, in stable and volatile periods, under MPH (purple) and PLA (green). (B) Learning rate for inferred-value learning, in stable and volatile periods, under MPH and PLA. (C) There was a significant interaction between drug and volatility for experienced-value learning, but not for inferred-value learning. (D) The interaction between drug and volatility was specific to experienced-value learning even for a sub-sample of participants who were keen inferred-value learners. Boxes = standard error of the mean, shaded region = standard deviation, individual datapoints are displayed, MPH = methylphenidate, PLA = placebo, * indicates statistical significance.

compared to PLA within stable (F(1,100) = 3.218, p < 0.076) but not volatile phases (F(1,100) = 1.387, p = 0.242). Simple effects analyses also revealed that, for experienced-value learning, learning rates were higher for volatile relative to stable phases under MPH ($\alpha_{\text{volatile\_experienced}}$ $\bar{x}(\sigma_{\bar{x}})$ = 0.353 (0.020), $\alpha_{\text{stable\_experienced}}$ $\bar{x}(\sigma_{\bar{x}})$ = 0.299(0.017), F(1,100) = 8.116, p = 0.005) but not PLA ($\alpha_{\text{volatile\_experienced}}$ $\bar{x}(\sigma_{\bar{x}})$ = 0.329(0.019), $\alpha_{\text{stable\_experienced}}$ $\bar{x}(\sigma_{\bar{x}})$ = 0.331(0.018), F(1,100) = 0.020, p = 0.888). There was no significant difference between volatile and stable learning rates for inferred-value learning under MPH ($\alpha_{\text{volatile\_inferred}}$ $\bar{x}(\sigma_{\bar{x}})$ = 0.187(0.013), $\alpha_{\text{stable\_inferred}}$ $\bar{x}(\sigma_{\bar{x}})$ = 0.214(0.015), simple main effect of volatility F(1,100) = 2.848, p = 0.095, or PLA ($\alpha_{\text{volatile\_inferred}}$ $\bar{x}(\sigma_{\bar{x}})$ = 0.187 (0.013), $\alpha_{\text{stable\_inferred}}$ $\bar{x}(\sigma_{\bar{x}})$ = 0.189(0.013), F(1,100) = 0.025, p = 0.876). Furthermore, as illustrated by the significant interaction, the difference between volatile and stable was greater under MPH (MPH difference $\bar{x}(\sigma_{\bar{x}})$ = 0.053(0.019), PLA difference $\bar{x}(\sigma_{\bar{x}})$ = − 0.002(0.016)).

There was no significant drug x volatility x learning type x group interaction (F(1,100) = 0.435, p = 0.511). Indeed, Bayesian statistics provided moderate support for the null hypothesis that the Social and Non-social groups did not differ (BF$_{01}$ = 4.66; see Appendix 3 for further analysis demonstrating an absence of group differences even in a subgroup that are particularly sensitive to the effects of MPH on inferred-value learning). There were no other significant interactions involving the factor drug (all p > 0.05). See *Appendix 3—table 2* for all other main effects and interactions. The interaction between drug, volatility and learning type remained significant (F(1,98) = 5.806, p = 0.018) when drug administration order (MPH Day 1, PLA Day 1) was included as a factor in the ANOVA. There was no drug x volatility x learning type x order interaction (F(1,98) = 0.203, p = 0.653) and no main effect of order (F(1,98) = 1.538, p = 0.218).

## MPH selectively affects experienced-value learning in 'keen inferred-value learners'

The red frame is a highly salient and behaviourally relevant cue. Optimal performance on this task requires participants to learn about the veracity of the frame and use this information to help choose amongst the boxes. Thus, the absence of a significant effect of MPH on inferred-value learning is particularly striking. Nevertheless, to guard against the possibility that the lack of a significant effect of MPH on inferred-value learning is driven by overall reduced reliance on inferred-value, relative to experienced-value, learning (i.e. that participants learn primarily from the blue/green boxes and ignore the frame) we ran the following post-hoc analysis. First, we identified whether experienced- or inferred-value learning was the primary driver of responses for each of our participants. This was achieved by regressing a Bayesian Learner Model (*Behrens et al., 2007*) of optimal responses against each participant's behaviour under PLA using the same method employed by *Cook et al. (2014)*. This produced two beta values for each participant representing the extent to which their responses were driven by experienced- ($\beta_{\text{experienced}}$) and inferred-value learning ($\beta_{\text{inferred}}$). 'Keen inferred-value learners' were defined as participants for whom $\beta_{\text{inferred}} > \beta_{\text{experienced}}$ (N = 45). Under PLA, these participants primarily use the indirect information (the frame) to make their decisions. In this subgroup, we observed the same significant drug x volatility x learning type interaction (F(1,44) = 12.186, p = 0.001; *Figure 3D*) as in the overall population. Under MPH $\alpha_{\text{volatile\_experienced}}$ ($\bar{x}(\sigma_{\bar{x}})$ = 0.367(0.027)) was significantly greater than $\alpha_{\text{stable\_experienced}}$ ($\bar{x}(\sigma_{\bar{x}})$ = 0.275(0.023); F(1,44) = 12.229, p = 0.001), there was no significant difference under PLA ($\alpha_{\text{volatile\_experienced}}$ $\bar{x}(\sigma_{\bar{x}})$ = 0.272(0.026), $\alpha_{\text{stable\_experienced}}$ $\bar{x}(\sigma_{\bar{x}})$ = 0.293(0.026), F(1,44) = 0.587, p = 0.448). Importantly, also in this subpopulation of keen inferred-value learners, for inferred-value learning the interaction between drug and volatility (F(1,44) = 2.418, p = 0.127, BF$_{01}$ = 3.25) and the main effect of drug (F(1,44) = 2.043, p = 0.160) were not statistically significant. Thus, we observed a selective effect of MPH on experienced-value learning even for participants whose responses under PLA were primarily driven by the advice.

## Optimal learner analysis

In the above section (*Learning rate analysis*), we report that the difference in $\alpha_{\text{volatile\_experienced}}$ and $\alpha_{\text{stable\_experienced}}$ is greater under MPH compared to PLA. To investigate whether, in this particular task, a greater difference between $\alpha_{\text{volatile\_experienced}}$ and $\alpha_{\text{stable\_experienced}}$ can be considered 'more optimal' we created an optimal learner model with the same architecture and priors as the adapted Rescorla-Wagner model employed above. By generating 100 synthetic datasets, which followed our probabilistic schedules, and fitting the optimal learner model to the synthetic data, we estimated

that the average optimal learning rates were $\alpha_{\text{optimal\_experienced\_stable}}$ = 0.16, $\alpha_{\text{optimal\_experienced\_volatile}}$ = 0.21, $\alpha_{\text{optimal\_inferred\_stable}}$ = 0.17, $\alpha_{\text{optimal\_inferred\_volatile}}$ = 0.19. Optimal values for $\zeta_{\text{optimal}}$ (the weight of inferred- relative to experienced-value learning) and $\beta_{\text{optimal}}$ were 0.23 and 1.44, respectively.

'Distance from optimal' indices (the difference between $\alpha$ and $\alpha_{\text{optimal}}$) were submitted to a 2 (drug: MPH, PLA) x 2 (volatility: volatile, stable) x 2 (learning type: experienced, inferred) x 2 (group: social, non-social) ANOVA. We observed a significant drug x learning type x volatility interaction (F (1,100) = 6.913, p = 0.010; *Figure 4*). Only for experienced-value learning was there a significant drug x volatility interaction (F(1,100) = 6.151, p = 0.015; inferred-value learning F(1,100) = 1.588, p = 0.211). Simple effects analyses demonstrated that experienced-value learning rates under PLA were further from the optimal value in the stable compared to volatile phase ($\alpha_{\text{volatile\_experienced-optimal}}$ $\bar{x}(\sigma_{\bar{x}})$ = 0.119(0.019), $\alpha_{\text{stable\_experienced-optimal}}$ $\bar{x}(\sigma_{\bar{x}})$ = 0.171(0.019), F(1,100) = 11.227, p = 0.001). The difference between volatile and stable was not significant under MPH ($\alpha_{\text{volatile\_experienced-optimal}}$ $\bar{x}(\sigma_{\bar{x}})$ = 0.143(0.020), $\alpha_{\text{stable\_experienced-optimal}}$ $\bar{x}(\sigma_{\bar{x}})$ = 0.139(0.017), F(1,100) = 0.051, p = 0.823). Numerically, stable learning rates were closer to optimal under MPH relative to PLA; however, this effect only approached statistical significance (F(1,100) = 3.218, p = 0.076). The distance from optimal for volatile learning rates was not significantly affected by MPH (F(1,100) = 1.387, p = 0.242). In sum, there was a trend such that under PLA participants systematically over-estimated the volatility of the stable environment and MPH counteracted this systematic bias.

## Relationship between accuracy scores and model-based analyses

Despite the absence of a significant drug effect on accuracy (see *Accuracy analysis*), we observed that experienced-value $\beta_{\text{WSLS\_vol-stable}}$ was significantly positively correlated with accuracy under

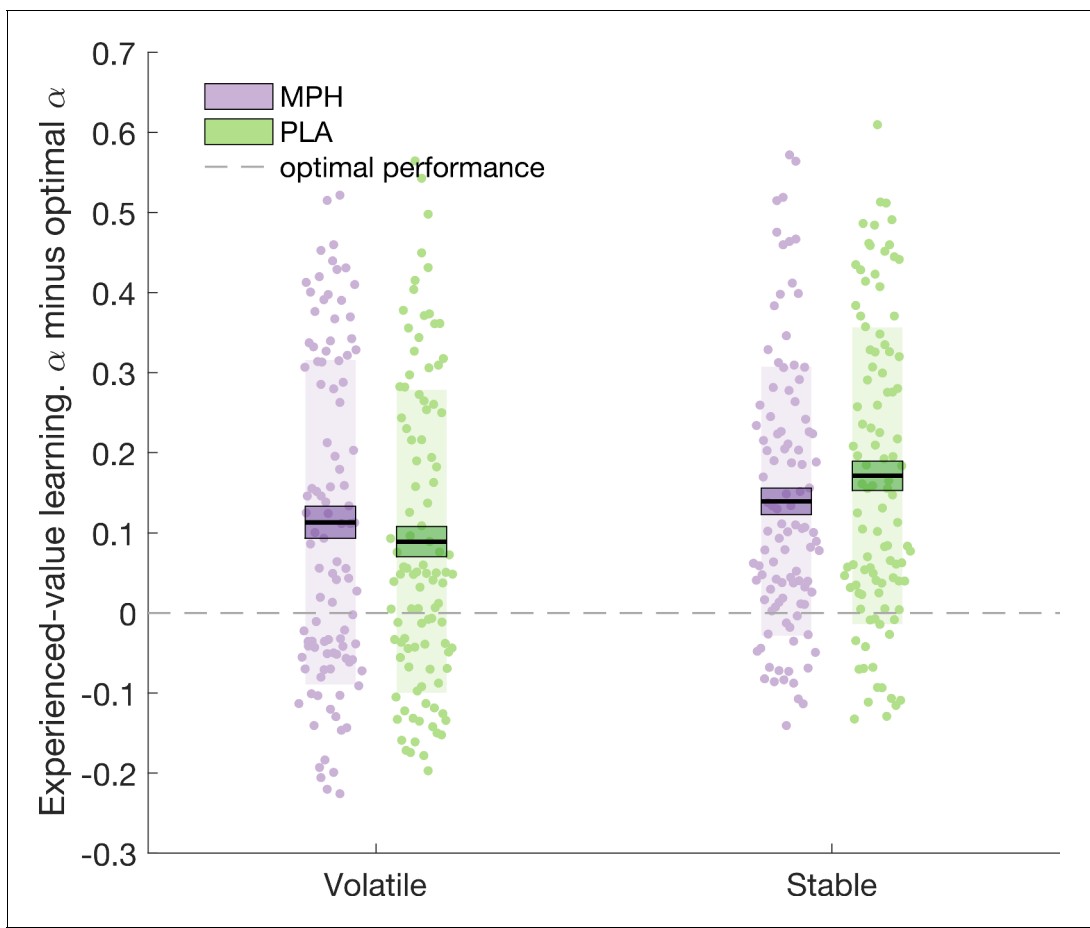

**Figure 4.** Experienced-value learning rates under PLA, but not MPH, were further from the optimal value in the stable compared to volatile phase.

both MPH (Pearson's r = 0.236, p = 0.017) and PLA (Pearson's r = 0.453, p < 0.001; see *Figure 2—figure supplement 1* for scatterplots). Thus, increased use of a WSLS strategy under volatile compared with stable phases was associated with better performance (more optimal choices) on this task.

The lack of a significant overall drug effect on accuracy was somewhat surprising given the significant (interaction) effect on learning rates. To investigate potential explanations, we conducted exploratory analyses concerning the relationship between estimated model parameters and accuracy. A backwards regression with MPH-PLA accuracy as dependent variable, and $\alpha_{\text{stable\_experienced\_MPH-PLA}}$, $\alpha_{\text{vol\_experienced\_MPH-PLA}}$, $\alpha_{\text{stable\_inferred\_MPH-PLA}}$, $\alpha_{\text{volatile\_inferred\_MPH-PLA}}$, $\beta_{\text{MPH-PLA}}$ and $\zeta_{\text{MPH-PLA}}$ as predictors, revealed that MPH-PLA accuracy was significantly predicted by the full model (R = 0.798, F(6,95) = 27.805, p < 0.001; *Table 1*). Removing predictors did not significantly improve the fit of the model ($R^2$change < 0.001, F change (1,95) = 0.030, p = 0.864). Including interaction terms for $\alpha_{\text{stable\_experienced\_MPH-PLA}}$ x $\alpha_{\text{vol\_experienced\_MPH-PLA}}$, and $\alpha_{\text{stable\_inferred\_MPH-PLA}}$ x $\alpha_{\text{vol\_inferred\_MPH-PLA}}$ did not significantly improve the fit of the model ($R^2$change = 0.003, F change (2,93) = 0.367, p = 0.694). Within the model the only significant predictors were $\alpha_{\text{volatile\_experienced\_MPH-PLA}}$, $\beta_{\text{MPH-PLA}}$, and $\zeta_{\text{MPH-PLA}}$ (*Table 1*).

Hierarchical regression showed that for $\zeta_{\text{MPH-PLA}}$ the $R^2$change value was 0.378 (F change (1,95) = 98.862, p <0.001) indicating that $\zeta_{\text{MPH-PLA}}$ accounts for an additional 38% of the variance compared to the other regressors in the model. The corresponding value for $\beta_{\text{MPH-PLA}}$ was 13% ($R^2$change = 0.127, F change (1,95) = 33.211, p <0.001) and 7% for $\alpha_{\text{volatile\_experienced\_MPH-PLA}}$ ($R^2$change = 0.073, F change (1,95) = 19.151, p <0.001). Hierarchical regression models conducted separately for PLA and MPH conditions also demonstrated that $\zeta$ scores are the best predictors of accuracy (Appendix 5).

Regressing model parameters against MPH-PLA accuracy demonstrates that any differences in accuracy scores between MPH and PLA conditions were predominantly driven by effects of MPH on $\zeta$ (38%) and, to a lesser extent, $\beta$ (13%) and $\alpha_{\text{volatile\_experienced}}$ (7%). In other words, participants who showed the greatest benefit of the drug in terms of accuracy scores were those who experienced the greatest decrease in $\zeta$ under MPH (where a decrease corresponds to a greater bias towards the use of experienced, as opposed to inferred, learned values). As noted above, there was no significant effect of drug on $\zeta$ across all participants ($\zeta_{\text{MPH}}$ $\bar{x}(\sigma_{\bar{x}})$ = 0.499(0.027), $\zeta_{\text{PLA}}$ $\bar{x}(\sigma_{\bar{x}})$ = 0.512(0.027); t (101) = -0.554, p = 0.581). The finding that changes in accuracy are accounted for by only a very small proportion of the variance in learning rate is consistent with the observation that the drug altered learning rate without altering accuracy.

## Summary of results

MPH increased participants' ability to adapt to environmental volatility, such that they exhibited increased learning rates (and increased use of a WSLS strategy) under volatile relative to stable conditions (*Figure 3*). This effect was only present in the case of direct, experienced-value learning. MPH did not modulate participants' ability to adapt to the volatility of the inferred utility of an indirect source of information. Although the drug significantly affected learning rate adaptation it did not affect accuracy, potentially because, for the current task, accuracy is predominantly influenced by the weighting of experience-value relative to inferred-value learning (i.e. $\zeta$ values).

**Table 1.** Coefficients from regression model with MPH-PLA accuracy as dependent variable.

| | $\beta$ | $\beta$(std. Err.) | standardised $\beta$ | t | p |
|---|---|---|---|---|---|
| *constant* | −0.008 | 0.006 | | −1.3 | 0.197 |
| $\beta_{\text{MPH-PLA}}$ | 0.014 | 0.002 | 0.36 | 5.763* | <0.001 |
| $\zeta_{\text{MPH-PLA}}$ | −0.241 | 0.024 | −0.658 | −9.943* | <0.001 |
| $\alpha_{\text{stable\_experienced\_MPH-PLA}}$ | 0.005 | 0.032 | 0.011 | 0.172 | 0.864 |
| $\alpha_{\text{vol\_experienced\_MPH-PLA}}$ | 0.126 | 0.029 | 0.285 | 4.376* | <0.001 |
| $\alpha_{\text{stable\_inferred\_MPH-PLA}}$ | 0.014 | 0.041 | 0.023 | 0.345 | 0.731 |
| $\alpha_{\text{volatile\_inferred\_MPH-PLA}}$ | 0.02 | 0.041 | 0.03 | 0.483 | 0.63 |

* indicates statistical significance.

## Discussion

Participants completed a learning task, requiring them to concurrently learn from their own direct experience, and from an additional indirect source of information. Participants completed the task once under PLA and once after MPH administration. Results confirmed the hypothesis that MPH, which blocks both noradrenaline and dopamine transporters, enhances participants' ability to adapt learning rate as a function of environmental change. This result advances the existing literature that, to date, has linked catecholamines to context-independent increases or decreases in learning rate, but not with volatility-based adjustments. In addition, we demonstrated that MPH only had a significant effect on direct, experienced-value learning: MPH had no significant effect on inferred-value learning and this was true irrespective of whether the indirect source was believed to be social or non-social in nature.

### MPH enhances learning rate adaptation as a function of environmental volatility

The current study established that MPH enhances the adaptation of experienced-value learning rates as a function of environmental volatility. Interestingly, we observed that under PLA, participants exhibited a bias towards over-estimating volatility (learning rates were generally greater than the corresponding optimal learner estimates). This bias was significantly greater for stable relative to volatile phases. MPH reduced learning rates in the stable context thereby bringing the stable learning rate closer to the optimal value. MPH did not bring learning rates closer to the optimal values in the volatile context. It is thus possible that, in line with a meta-learning account, MPH optimises learning rate but that, across the group, this effect could only be observed in the stable context where learning rates were especially non-optimal.

Our optimal learner analysis also demonstrated that for optimal performance $\alpha_{\text{volatile}}$ should be greater than $\alpha_{\text{stable}}$. Consequently, it is striking that under PLA our participants did not exhibit a significant difference between $\alpha_{\text{volatile}}$ and $\alpha_{\text{stable}}$. By fitting learning models to participants' responses, previous studies have demonstrated higher learning rates in volatile compared to stable phases (e.g. *Browning et al., 2015*; *Diaconescu et al., 2014*). However, such studies, differ from ours in that demands for learning were lower than those in the current study: Participants learned from only one source of information (*Browning et al., 2015*) or were provided with explicit outcome probabilities (*Diaconescu et al., 2014*). To the best of our knowledge, the current experiment is the first to estimate learning rates, for volatile and stable phases, when participants are simultaneously learning from two sources of information. Here, we demonstrated a lack of adaptation to the volatility of the environment under PLA, (possibly due to high demands for learning from dual sources), which is rescued by MPH.

Our results comprise an important step in understanding the neurochemical drivers of learning rate adjustment. Potential exact neurobiological mechanisms include meta-learning models which suggest a critical role for phasic dopamine in training the dynamics of a prefrontal system, which then learns about the structure of the environment (*Collins and Frank, 2016*; *Massi et al., 2018*; *Pasupathy and Miller, 2005*; *Wang et al., 2018*). The implication of such models is that better learning of the structure of the environment, for example current levels of volatility, results in more adaptive learning rate adjustment. In terms of neurobiological implementation, by blocking dopamine reuptake, MPH likely prolongs the impact of phasic dopamine signals (*Volkow et al., 2002*). This could then potentially enhance learning rate adjustment by improving learning about the structure of the environment.

A different theory suggests that learning rate is optimised based on a formal comparison between medium- and long-term running averages of reward rate (*Doya, 2002*; *Schweighofer and Doya, 2003*), carried by the phasic and tonic firing of dopamine neurons respectively (*Schweighofer and Doya, 2003*). According to this model, learning rates are low when the difference between phasic and tonic signals is small, and high when the difference is large. Typically, phasic bursting has negligible effects on tonic dopamine levels (*Floresco et al., 2003*), thus the two are relatively independent: tonic firing can be high whilst phasic is low or vice versa. However, under MPH, dopamine reuptake is inhibited thus phasic bursting likely increases tonic levels via dopamine overflow from the synaptic cleft (*Grace, 1991*). Recent evidence also suggests that phasic bursting can promote dopamine transporter internalisation

(*Lohani et al., 2018*), thus further creating dependency, and potentially reducing differences, between tonic and phasic rates. Consequently, we speculate that MPH may increase the dependency between phasic and tonic dopamine levels. Whilst this would have negligible effects on the difference between phasic and tonic levels in reward-poor (potentially volatile) environments, in reward-rich (potentially stable) environments increases in phasic signals would result in higher tonic levels and thus low learning rates.

Evidence from studies linking dopamine with the modulation of learning rates (*Marshall et al., 2016*), or individual differences therein (*Krugel et al., 2009*; *Set et al., 2014*), supports the hypothesis that MPH might adapt learning rates via modulating dopamine. However, it is also plausible that the effects we observed reflect MPH's influence on the noradrenergic system. Indeed, studies that have tried to disentangle the roles of dopamine and noradrenaline in complex learning tasks have argued that, though dopamine is linked to 'lower level' modulation of motor responses as a function of learned probabilities, noradrenaline is associated with 'higher level' learning about environmental changes such as changes in volatility (*Marshall et al., 2016*). Indeed, previous work supports a role for noradrenaline in learning rate adaptation. Theoretical and modelling work has suggested that noradrenaline acts as a 'neural interrupt signal' which alerts the learner to an unpredicted change in the learning environment (*Dayan and Yu, 2006*; *Yu and Dayan, 2005*) and performs a 'network reset' (*Bouret and Sara, 2005*). In other words, noradrenaline is important in 'change detection'. In line with this, pharmacological manipulations and lesions of the noradrenergic system affect performance on tasks such as reversal learning and attentional-set shifting which require detection of, and adaptation to, environmental changes (*Devauges and Sara, 1990*; *Lapiz and Morilak, 2006*; *Lapiz et al., 2007*; *McGaughy et al., 2008*; *Newman et al., 2008*; *Seu et al., 2009*; *Tait et al., 2007*). In humans, uncertainty about environmental states (*Muller et al., 2019*), learning rates (*Browning et al., 2015*; *Nassar et al., 2012*; *Silvetti et al., 2013*) and prediction errors (*Lavín et al., 2013*; *Preuschoff et al., 2011*) are correlated with pupil size, often considered an indirect index of locus coeruleus (the main noradrenergic nucleus of the brain) activity (*Aston-Jones and Cohen, 2005*; *Gilzenrat et al., 2010*; *Joshi et al., 2016*; *Murphy et al., 2014*), and are modulated by pharmacological manipulation of the noradrenaline system (*Jepma et al., 2016*; *Marshall et al., 2016*). In the context of our task, the 'unpredicted change in learning environment' corresponds to a switch from stable to volatile phase. By administering MPH, we may have increased participants' synaptic noradrenaline levels (*Kuczenski and Segal, 1997*) and reduced spontaneous activity in locus coeruleus neurons via noradrenaline action at α-adrenergic receptors on locus coeruleus cells (*Devilbiss and Berridge, 2008*). Thus, perhaps MPH shifted locus coeruleus activity toward a phasic mode, in which cells respond more strongly (and specifically) to salient events (*Aston-Jones et al., 2007*), such as the switch from stable to volatile environments.

The potential neurobiological mechanisms described above are not mutually exclusive. It is possible that, via effects on both dopaminergic and noradrenergic systems, MPH enhances a) learning about environmental structure, b) learning rate optimisation and c) change detection, and that these three processes work in concert. Active debate, however, highlights potential redundancies: *Farashahi et al. (2017)*, for example, argue that adjustment of learning rate can be achieved in the absence of explicit optimisation or learning of environmental structure, via dopamine-dependent meta-plasticity at the level of the synapse (*Moussawi et al., 2009*; *Yagishita et al., 2014*). Indeed, Farashahi and colleagues demonstrate that their biophysically inspired model, which adjusts learning parameters via metaplastic synapses that can occupy states with different levels of synaptic efficacy, accurately predicts learning rate adjustment as a function of volatility. Whilst this debate remains unresolved, it is possible that MPH affects multiple processes which all contribute to the adjustment of learning rate as a function of environmental volatility.

Furthermore, it should be acknowledged that other neuromodulators, for example acetylcholine, have also been linked with change detection and volatility processing (*Yu and Dayan, 2005*; *Marshall et al., 2016*). In the current study, a single pharmacological agent was employed. Thus, our results may, in principle, reflect general effects of neuromodulation. Using a design, such as that employed by *Marshall et al. (2016)*, who compared the effects of a noradrenergic modulator (prazosin), a cholinergic modulator (biperiden) and a dopaminergic modulator (haloperidol), would enable us to make neuromodulator-specific claims. Indeed, based on this comparison, Marshall and

colleagues propose that acetylcholine guides probabilistic learning *within* environmental contexts, while noradrenaline has a more circumscribed role in learning about differences in volatility *across* different environmental contexts.

## MPH significantly affects experienced-, but not inferred-, value learning

The absence of a significant effect of MPH on inferred-value learning is particularly striking given that a) the red frame is a highly salient, behaviourally relevant, attentional cue, and b) meta-learning theories do not, in principle, predict differential effects of catecholaminergic modulation on experienced versus inferred-value learning. Why would MPH significantly affect experienced-, but not inferred-value learning? One possibility is that our participants simply ignored the information that was represented by the frame, in other words, that the current paradigm is poorly suited to detect variation in inferred-value learning. Refuting this, we have demonstrated that MPH significantly affects experienced-value learning *even for participants whose responses are primarily driven by inferred-value learning* (see results section 'MPH selectively affects experienced-value learning in keen inferred-value learners'). Thus, the lack of significant effect of MPH on inferred-value learning is not driven by a tendency to ignore the frame.

Another potential explanation for the lack of significant effects of MPH on inferred-value learning relates to the complex effects of MPH on working memory. *Fallon et al. (2017)* have demonstrated that MPH improves distractor resistance while impairing working memory updating. More specifically, in a delay phase between initial presentation of to-be-remembered information and recall, Fallon and colleagues presented additional information which participants should sometimes ignore and sometimes use to replace (update) the original to-be-remembered information. MPH improved ignoring, but not updating - an effect that Fallon and colleagues interpret in the context of dual state theory (*Durstewitz and Seamans, 2008*). According to this theory, intermediate levels of dopamine favour stabilisation of representations, whereas excessively high or low levels of dopamine favour representation destabilisation. The current task can be conceptualised in a similar fashion to Fallon and colleagues' paradigm. Participants first see the direct source (blue and green boxes), providing the opportunity to reflect on the history of outcomes and estimate the value associated with each box. In the 'delay phase' between seeing the boxes and making their response, participants see the indirect source (frame). By reflecting on the history of the veracity of the frame, they can estimate the probability that the frame provides correct advice, and decide whether to *update* their choice or *ignore* the frame. Consequently, because the direct source (boxes) always came first, and the indirect source (frame) second, positive effects of MPH on ignoring, not updating (as shown by *Fallon et al., 2017*), would only benefit experienced-value, not inferred-value learning. Such positive effects of MPH on ignoring may improve experienced-value learning by making it more robust against misleading information from the frame.

## Comparable effects of MPH for social and non-social groups

It is notable that our results do not provide evidence in support of a dissociation between social and non-social learning. Note that the lack of a difference between social and non-social groups cannot be due to poor comprehension of the cover story: Participants were prohibited from starting the task until they scored 100% correct in a quiz which included questions about the nature of the red frame (see Materials and methods: learning task). Furthermore, using the same task, we have previously demonstrated that social dominance predicts learning for participants that are given the social, but not the non-social, cover story (*Cook et al., 2014*). Thus, belief about the origin of the information represented by the frame modulates the way in which this information is processed. The absence of an effect of MPH on inferred-value learning from an additional source, even when the indirect source is believed to be social in nature, is perhaps surprising given positive effects of MPH on social influence (*Campbell-Meiklejohn et al., 2012*), reports that social learning activates catecholamine rich areas of the brain such as the striatum (*Biele et al., 2009*; *Braams et al., 2014*; *Campbell-Meiklejohn et al., 2010*; *Delgado et al., 2005*; *Diaconescu et al., 2017*; *Klucharev et al., 2009*), and evidence that ventral striatal prediction errors during social learning are influenced by a gene involved in regulating dopamine degradation (*Diaconescu et al., 2017*). An important consideration, however, is that in studies that have linked social learning to signals in catecholamine-rich areas of the brain, the social

information is not an *additional* source to be learned about in concert with learning from one's own experience, rather, the social source is the *only* source that participants must learn from. In other words, in existing studies the social information is typically not 'inferred', and is therefore more comparable to our experienced-value condition.

### Conclusion

We show that relative to placebo, methylphenidate modulates the ability to adapt learning rate from a direct source of information but does not modulate learning from an indirect, inferred, source. This observation was consistent when indirect information was from either a social or non-social source. These results advance existing literature by documenting a causal link between catecholaminergic modulation and the adjustment of learning rate as a function of environmental volatility.

## Materials and methods

The general study design is identical to that reported in *Swart et al. (2017)* and *Fröböse et al. (2018)* and is included here for completeness.

### General procedure and pharmacological manipulation

The study consisted of two test sessions with an interval of one week to 2 months. The first test day started with informed consent, followed by medical screening. Participation was discontinued if participants met any of the exclusion criteria. On both test days, participants first completed baseline measures. Due to previous work demonstrating baseline-dependent effects of MPH, we included two assessments (Listening Span Test: *Daneman and Carpenter, 1980*; *Salthouse and Babcock, 1991*; Barratt Impulsiveness Scale (BIS-11): *Patton et al., 1995*) that have been demonstrated, with positron emission tomography, to relate to dopamine baseline function (*Cools et al., 2008*; *Landau et al., 2009*; *Buckholtz et al., 2010*; *Kim et al., 2014*; *Reeves et al., 2012*). Next participants received a capsule containing either 20 mg MPH (Ritalin, Novartis) or PLA, in a double-blind, placebo-controlled, cross-over design. MPH blocks the dopamine and noradrenaline transporters, thereby diminishing the reuptake of catecholamines. When administered orally, MPH has a maximal plasma concentration after 2 hr and a plasma half-life of 2–3 hr (*Kimko et al., 1999*). After an interval of 50 min, participants started with the task battery containing the learning task preceded by a task adapted from *Geurts et al. (2013)* and succeeded by tasks reported in *Swart et al. (2017)*, *Fröböse et al. (2018)* and adapted from *Fallon et al. (2017)*. On average, the learning task reported here was performed 1 hr 20 min after capsule intake, well within the peak of plasma concentration. Both test days lasted approximately 4.5 hr, which participants started at the same time (maximum difference of 45 min). Blood pressure, mood and potential medical symptoms were monitored thrice each day: before capsule intake, upon start of the task battery and after finishing the task battery. Participants were asked to abstain from alcohol and recreational drugs 24 hr prior to testing and from smoking and drinking coffee on the days of testing. Participants completed self-report questionnaires at home between (but not on) test days. Upon completion of the study, participants received monetary reimbursement or study credits for participation. The study was in line with the local ethical guidelines approved by the local ethics committee (CMO protocol NL47166.091.13; trial register NTR4653) and in accordance with the Helsinki Declaration of 1975. Baseline measures, baseline-dependent effects of MPH, self-report questionnaires, mood-ratings and medical symptoms are reported in Appendix 6.

### Participants

Data was collected from 106 native Dutch volunteers (aged 18–28 years, $\bar{x}(\sigma)$ = 21.5 (2.3); 53 women; 84 right-handed; sample size calculation reported in CMO registered protocol NL47166.091.13). Four participants dropped out after the first test day (due to excessive delay between test days, loss of motivation, nausea, and mild arrhythmia). Of the resulting 102 participants, 50 participants received MPH on the first day. Exclusion criteria comprised a history of psychiatric, neurological or endocrine disorders. Further exclusion criteria were autonomic failure, hepatic, cardiac, obstructive respiratory, renal, cerebrovascular, metabolic, ocular or pulmonary disease, epilepsy, substance abuse, suicidality, hyper/hypotension, diabetes, pregnancy/breastfeeding, lactose

intolerance, regular use of corticosteroids, use of psychotropic medication or recreational drugs 24 hr before each test day, and first degree family members with schizophrenia, bipolar disorder, ventricular arrhythmia or sudden death. Excluded volunteers received €10,- as a compensation for their time and effort.

## Learning task

Participants completed a modified version (*Cook et al., 2014*) of a probabilistic learning task developed by *Behrens et al. (2008)*. The task was programmed using the Psychophysics toolbox (*Brainard, 1997*; *Pelli, 1997*) in Matlab R2014b (The MathWorks, Natick, MA; RRID:SCR_001622). On each trial, participants made a choice between a blue and a green box. Correct choices were rewarded with points represented on a bar spanning the bottom of the screen. Participants' aim was to accumulate points to obtain a silver (€2) or gold (€4) reward. Throughout the experiment, the probability of reward associated with each box (blue or green) varied according to a probabilistic schedule. Participants were informed that the task followed 'phases' wherein sometimes blue, but at other times green, would be more likely to result in reward.

In addition to the reward history, a second source of information was available to participants. On each trial, before participants made their choice, a frame appeared. Participants in the Social Group were instructed that this frame represented choices made by a group of participants who had completed the task previously. Participants in the Non-social Group were instructed that the frame represented the outcome from a system of rigged roulette wheels. Both groups were informed that the frame would fluctuate between providing predominantly correct and predominantly uninformative 'advice' (see Appendix 2 for instruction scripts).

Participants in both the Social and Non-social Groups were pseudorandomly allocated to one of four different randomisation groups, group membership determined the probabilistic schedule underpinning outcomes (blue/green) and the veracity of the frame (correct/incorrect). These were based on the schedules used by *Behrens et al. (2008)*. However, the schedules were counterbalanced between participants to ensure that superior learning of direct over indirect information could not be explained in terms of superior learning of schedules with increased/decreased, or early/late occurring, volatility (*Figure 1—figure supplement 1*). Although probabilistic schedules for Day 2 were the same as Day 1, there was variation in the trial-by-trial outcomes and advice. To prevent participants from transferring learned stimulus-reward associations from Day 1 to Day 2 different stimuli were also employed: 50% of participants viewed yellow/red triangles with advice represented as a purple frame on Day 1 and blue/green squares with advice as a red frame on Day 2, for 50% of participants this order was switched. Before participation all participants completed a self-paced step-by-step on-screen explanation of the task in which they first learned to choose between two options to obtain a reward, and subsequently learned that the advice represented by the frame could help them in making an informed decision about their choice. Following the task explanation, participants were required to complete a short quiz testing their knowledge of the task. Participants were required to repeat the task explanation until they achieved 100% correct score in this quiz. Consequently, we could be sure that all participants understood the structure of the task and were aware that the red/purple frame represented either social information (Social Group), or information from a rigged set of roulette wheels (Non-social Group).

All participants completed 120 trials on Day 1 and Day 2. The task lasted approximately 25 min, including instructions. At the end of the task, participants were informed whether they had scored enough points to obtain a silver/gold award.

## Statistical analyses

Analyses were conducted using Matlab R2014b (The MathWorks, Natick, MA) and IBM SPSS Statistics version 22 (IBM Corp., Armonk, NY). Bayesian analyses were run in JASP (*JASP Team, 2018*). All raw data and analysis scripts can be accessed here: https://osf.io/z59us.

## Win-stay, lose-shift analysis

WSLS behaviour was quantified using a logistic regression on individual participant choices (green/blue; follow/deviate from advice), to assess the degree to which a WSLS strategy guided these choices. For each participant, for each of the eight cells of the 2x2x2 factorial design (drug x

volatility x learning type), a regressor was created representing the choice that they would have made, for each trial, if they strictly adhered to a WSLS strategy. For experienced-value learning, if on the previous trial the participant had chosen blue and won, the regressor would predict that the participant should *stay* with a blue choice (coded as 1); if, however, the participant had chosen blue and lost, the regressor would predict a *shift* to a green choice (coded as 0). Similarly, for inferred-value learning, if the participant had previously followed the frame's advice and lost, the regressor would predict that the participant should *shift* such that they do not follow the frame's advice in the current trial. Regressing these predictors against each participant's actual choices, in a series of 8 separate univariate regressions, provided 8 beta values ($\beta_{WSLS}$) per participant where high values indicate that a WSLS strategy accounts for a large amount of variance in the participant's choices. $\beta_{WSLS}$ were entered into a single repeated measures ANOVA with within-subjects factors drug (MPH, PLA), volatility (stable, volatile), learning type (experienced, inferred) and between-subjects factor group (social, non-social).

## Learning rate analysis

Learning rates for stable and volatile phases were estimated by fitting a simple learning model to participants' choice data. This learning model consisted of two Rescorla-Wagner predictors (*Rescorla and Wagner, 1972*) (one which estimated the utility of a blue choice ($r_{\_direct(i+1)}$) and one which estimated the utility of going with the red/purple frame ($r_{\_indirect(i+1)}$)) coupled to an action selector (*Browning et al., 2015*; *Behrens et al., 2007*). Model fitting was performed using scripts adapted from the TAPAS toolbox (http://www.translationalneuromodeling.org/tapas), study-specific scripts are available at https://osf.io/z59us. Data were initially analysed with three different models: Model 1 comprised a classic Rescorla-Wagner model, Model 2 extended Model 1 with separate learning rates $\alpha$ for volatile and stable blocks and Model 3 extended Model 2, with (four) separate learning rates for stable and volatile blocks for both experienced- and inferred-value learning. Bayesian model selection (*Daunizeau et al., 2014*) - where log model evidence was approximated with F values - demonstrated that Model 3 provided the best fit to the data (Appendix 4).

## Rescorla-Wagner predictors

Our Rescorla-Wagner predictors comprised modified versions of a simple delta-learning rule (*Rescorla and Wagner, 1972*). This rule typically has a single free parameter, the learning rate ($\alpha$) and estimates outcome probabilities using *Equation (1)*.

$$V_{(i+1)} = V_i + \alpha(r_i - V_i) \tag{1}$$

where $V_{(i+1)}$ is the predicted value (probability of reward) for ($i$+1)th trial, $r_i$ is reward received on the $i$th trial, and the prediction error ($r_i - V_i$) is used to update the stimulus value, using the learning rate $\alpha$. Different values for $\alpha$ will result in different learnt values of that stimulus. Our modified version separately estimated $\alpha$ for stable and volatile phases. Furthermore, we simultaneously ran two Rescorla-Wagner predictors such that we could estimate $\alpha$ values relating to both experienced- and inferred-value learning. Subsequently, our model generated the predicted value of a blue choice based on previous experience ($V_{\_experienced(i+1)}$) and of the choice to go with the red/purple frame ($V_{\_inferred(i+1)}$) based on inference about whether the frame's recommended choice has been reliably the same as the choice that the participant has reliably experienced to be correct, and provided four $\alpha$ estimates: $\alpha_{stable\_experienced}$, $\alpha_{volatile\_experienced}$, $\alpha_{stable\_inferred}$, $\alpha_{volatile\_inferred}$.

### Action selector

As participants had access to both information from experienced-value learning and inferred-value learning, our response model assumed that participants integrated both sources of information in order to predict the probability that a blue choice would be rewarded. Based on work by *Diaconescu et al. (2014)* and *Diaconescu et al. (2017)*, we calculated the estimated value of a blue choice according to *Equation (2)*, where we combine the value of inferred advice and the value of experienced information:

$$vB_{(i+1)} = \zeta\left(V_{\_inferred\_advice\_weighted(i+1)}\right) + (1-\zeta)\left(V_{\_experienced(i+1)}\right) \tag{2}$$

In *Equation (2)*, ζ is a parameter that varies between individuals and which controls the weighting of indirect relative to direct sources of information. $V_{\_inferred\_advice\_weighted(i+1)}$ comprises the advice provided by the frame weighted by the probability of advice accuracy ($V_{\_inferred(i+1)}$) in the context of making a blue choice (*Equation (3)*).

$$V_{\_inferred\_advice\_weighted(i+1)} = \left|\text{advice} - V_{\_inferred(i+1)}\right| \tag{3}$$

where advice = 0 for blue and advice = 1 for green. Thus, if the frame advised 'green' and the probability of advice accuracy was estimated at 80% ($V_{\_inferred(i+1)}$ = 0.80), the probability of a blue choice being rewarded, inferred from inferred-value learning, would be 0.2 ($V_{\_inferred\_advice\_weighted(i+1)}$ = $|1 - 0.8| = 0.2$). Conversely the probability, inferred from inferred-value learning, that the correct choice is green would equal 0.8.

The probability that the participant follows this integrated belief ($vB_{(i+1)}$), was described by a unit square sigmoid function (*Equation (4)*); here, responses are coded as $y_{(i+1)} = 1$ when selecting the blue option, and $y_{(i+1)} = 0$ when selecting green.

$$P\left(y_{(i+1)} = 1 \,\|\, vB_{(i+1)}\right) = \frac{vB^{\beta}_{(i+1)}}{vB^{\beta}_{(i+1)} + \left(1 - vB_{(i+1)}\right)^{\beta}} \tag{4}$$

β is a participant-specific free parameter representing the inverse of the decision temperature: as β→∞, the sigmoid function approaches a step function with a unit step at $vB_{(i+1)}$ = 0.5 (i.e. no decision noise).

## Model fitting

Optimisation was performed using a quasi-Newton optimisation algorithm (*Broyden, 1970*; *Fletcher, 1970*; *Goldfarb, 1970*; *Shanno, 1970*) as specified in quasinewton_optim_config.m. The objective function for maximization was the log-joint posterior density over all parameters, given the data and the generative model. $\alpha$ values were estimated in logit space (see tapas_logit.m) where 'Logit-space' is a logistic sigmoid transformation of native space (tapas_logit(x) = ln(x/(1-x)); x = 1/(1 +exp(-tapas_logit(x)))). The prior for $\alpha$ was tapas_logit(0.2, 1), with a variance of 1. Which was chosen to be uninformative and allow for substantial individual differences in learning. Initial values were fixed at tapas_logit(0.5, 1). The prior for $\beta$ was log(48), with a variance of 1, and the prior for ζ which controls the weighting between direct and indirect sources of information was set at 0 with a variance of 102 (logit space) which corresponds to an equal weighting for information derived from experienced-value and inferred-value learning. Priors were selected based on pilot data and previous work. Given the priors over parameters and the input sequence, maximum-a-posteriori (MAP) estimates of model parameters were calculated using the HGF toolbox version 3. The code used is freely available as part of the open source software package TAPAS at http://www.translationalneuromodeling.org/tapas.

## Model validation

An illustration of the ability of the model to recapitulate participant responses is provided in *Figures 5* and *6* below, which juxtapose participants' accuracy and choices (running average, across five trials) next to accuracy and choices derived from model simulations (using the model's parameter estimates to simulate choices according to tapas_simModel.m). For illustrative purposes, we present data from the four different randomisation schedules separately. To more formally test the model's predictions of participant behaviour, we calculated, for each participant, the average value that the model estimated for the options chosen by the participant (collapsed across days 1 and 2), and the average value that the model estimated for the options that were not chosen by the participant. If the model accurately describes participants' choice behaviour, then average values for chosen options should be significantly higher than for unchosen options. Indeed, a paired samples t-test illustrated that, model-derived values for chosen options ($\bar{x}(\sigma)$ = 0.628(0.077)) were significantly greater than values for unchosen options ($\bar{x}(\sigma)$ = 0.381 (0.086); $t(101)$ = 22.524, p < 0.001).

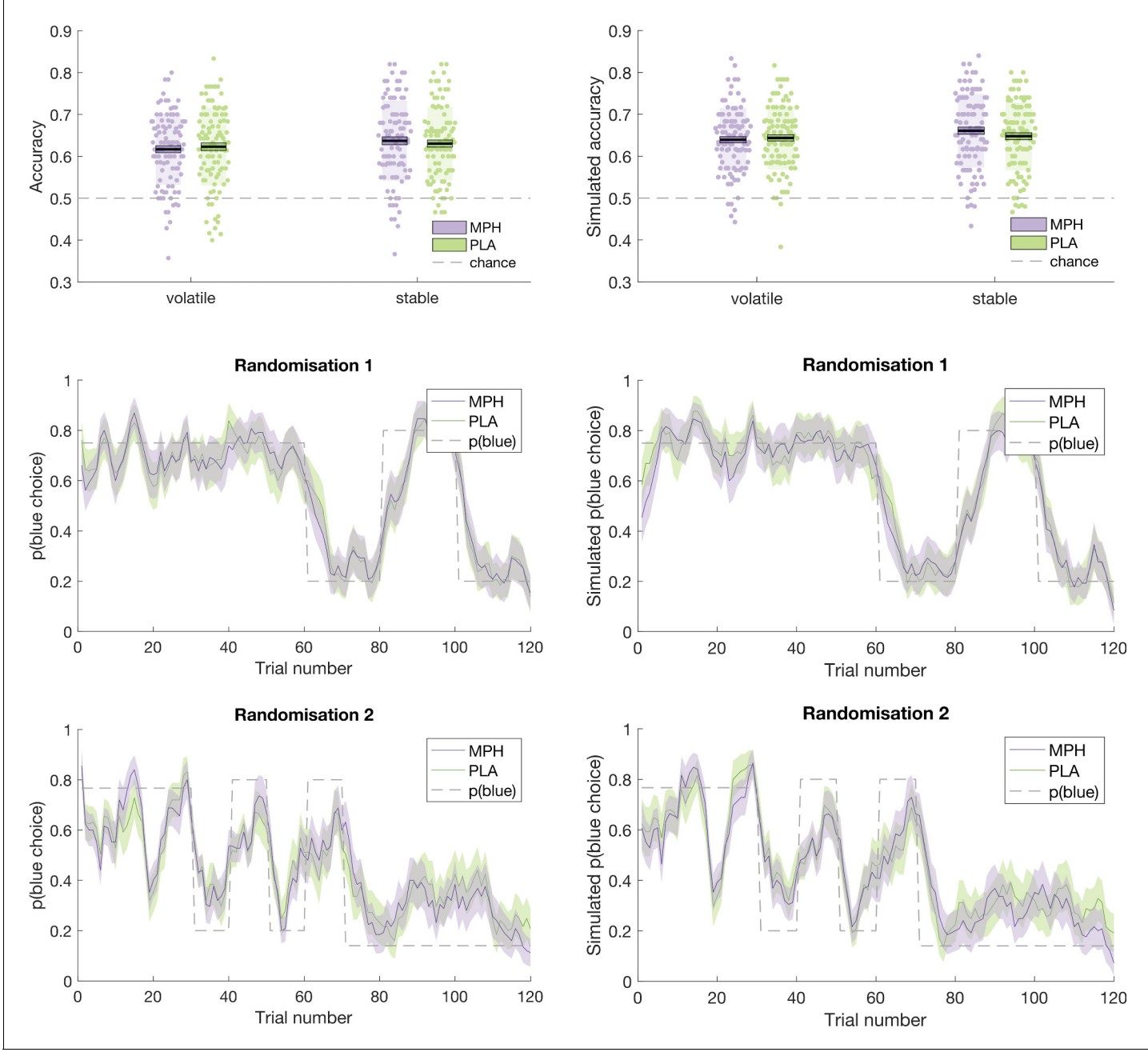

**Figure 5.** Participant data (left) juxtaposed against model simulations (right). Top: Accuracy, in stable and volatile phases, under MPH (purple) and PLA (green). Boxes = standard error of the mean, shaded region = standard deviation, individual datapoints are displayed. Middle and Bottom: Running average, across 5 trials, of blue choices for probabilistic randomisation schedules 1 (middle) and 2 (bottom). Shaded region = standard error of the mean. MPH = methylphenidate, PLA = placebo.

## Optimal learner analysis

To calculate the optimal parameters for the above described adapted Rescorla-Wagner model, 100 synthetic datasets were created, each of which followed one of the four pre-programmed probabilistic schedules. An optimal learner model, with the same architecture and priors as the adapted Rescorla-Wagner model, was fit to each synthetic dataset. Parameters estimates were averaged over the 100 model fits to synthetic data.

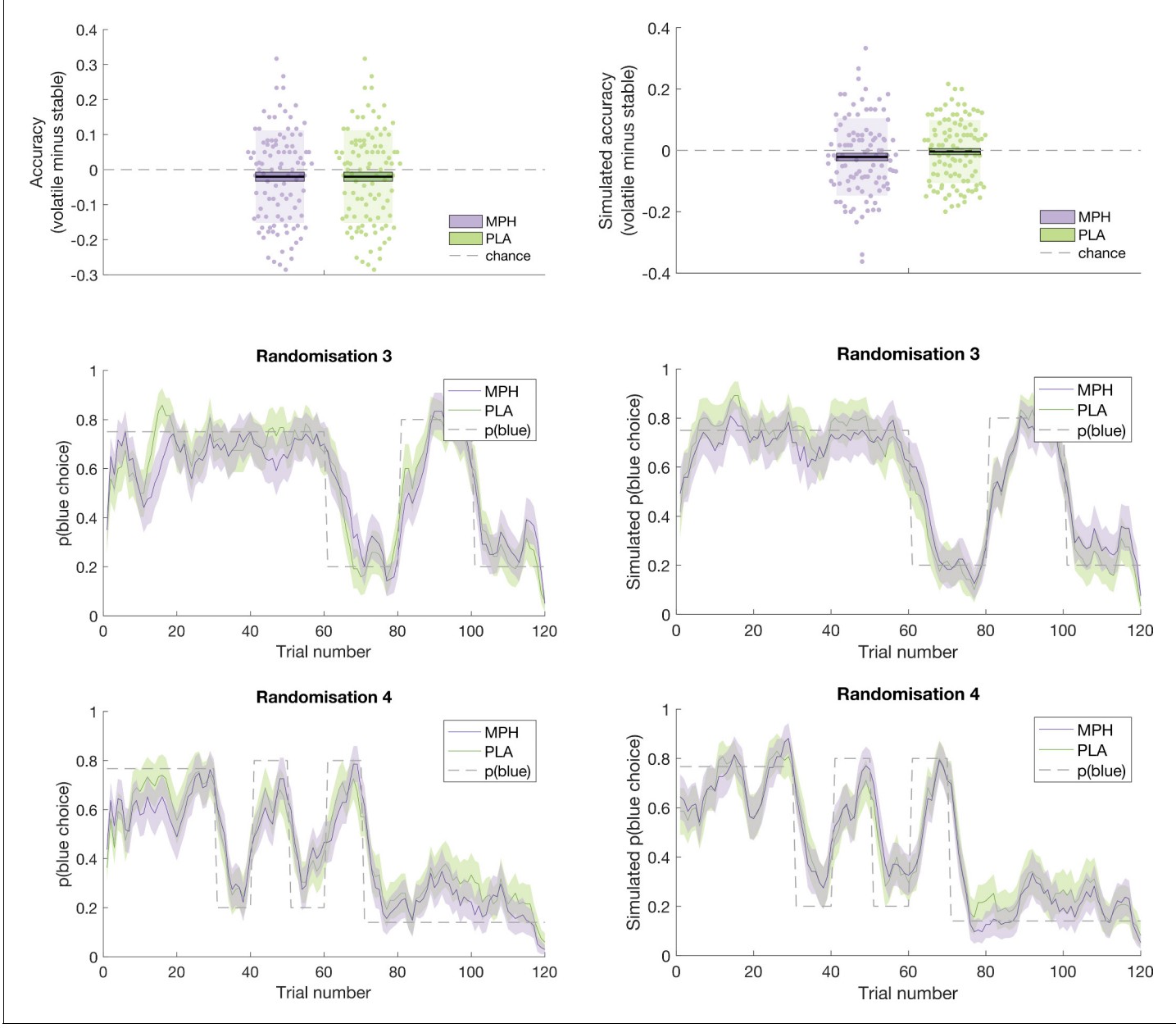

**Figure 6.** Participant data (left) juxtaposed against model simulations (right). Top: Accuracy, in stable and volatile phases, under MPH (purple) and PLA (green). Boxes = standard error of the mean, shaded region = standard deviation, individual datapoints are displayed. Middle and Bottom: Running average, across 5 trials, of blue choices for probabilistic randomisation schedules 3 (middle) and 4 (bottom). Shaded region = standard error of the mean. MPH = methylphenidate, PLA = placebo.

## Acknowledgements

We thank Dr Monique Timmer and Dr Peter Mulders for medical assistance, Dr Sean James Fallon for advice on setting up the MPH study, Mr Alex Chamberlain for help with data collection, Dr Eliana Vassena and for comments on a previous draft and Dr Rebecca Lawson for sharing niceGroupPlot.m.

# Additional information

## Competing interests

Hanneke EM den Ouden: has acted as consultant for Eleusis benefit corps but does not own shares. Eleusis have no involvement in this study. Roshan Cools: has acted as a consultant for Pfizer and Abbvie but does not own shares. Pfizer and Abbvie have no involvement in this study. The other authors declare that no competing interests exist.

## Funding

| Funder | Grant reference number | Author |
|---|---|---|
| H2020 European Research Council | ERC Starting Grant 757583 | Jennifer L Cook |
| Nederlandse Organisatie voor Wetenschappelijk Onderzoek | Research talent grant 406-14-028 | Jennifer C Swart |
| University of Birmingham | Birmingham Fellows Programme | Jennifer L Cook |
| ZonMw | 92003576 | Dirk EM Geurts |
| James S. McDonnell Foundation | James McDonnell scholar award | Roshan Cools |
| Nederlandse Organisatie voor Wetenschappelijk Onderzoek | Vici Award 453-14-005 | Roshan Cools |
| Nederlandse Organisatie voor Wetenschappelijk Onderzoek | Veni Grant 451-11-004 | Hanneke EM den Ouden |
| Swiss National Science Foundation | PZ00P3_167952 | Andreea O Diaconescu |
| Krembil Foundation | | Andreea O Diaconescu |

The funders had no role in study design, data collection and interpretation, or the decision to submit the work for publication.

## Author contributions

Jennifer L Cook, Conceptualization, Resources, Data curation, Formal analysis, Investigation, Visualization, Methodology, Project administration; Jennifer C Swart, Monja I Froböse, Conceptualization, Data curation, Formal analysis, Methodology, Project administration; Andreea O Diaconescu, Conceptualization, Resources, Data curation, Formal analysis, Methodology, Project administration; Dirk EM Geurts, Conceptualization, Data curation, Methodology, Project administration; Hanneke EM den Ouden, Conceptualization, Data curation, Supervision, Investigation, Methodology, Project administration; Roshan Cools, Conceptualization, Resources, Data curation, Supervision, Funding acquisition, Investigation, Methodology, Project administration

## Author ORCIDs

Jennifer L Cook (iD) https://orcid.org/0000-0003-4916-8667
Jennifer C Swart (iD) http://orcid.org/0000-0003-0989-332X
Andreea O Diaconescu (iD) https://orcid.org/0000-0002-3633-9757

## Ethics

Human subjects: Human subjects: Informed consent, and consent to publish, was obtained prior to participation. The study was in line with the local ethical guidelines approved by the local ethics committee (CMO / METC Arnhem Nijmegen: protocol NL47166.091.13), pre-registered (trial register NTR4653, http://www.trialregister.nl/trialreg/admin/rctview.asp?TC=4653), and in accordance with the Helsinki Declaration of 1975.

Decision letter and Author response
Decision letter https://doi.org/10.7554/eLife.51439.sa1
Author response https://doi.org/10.7554/eLife.51439.sa2

## Additional files

### Supplementary files
• Transparent reporting form

### Data availability
All raw data and analysis scripts can be accessed at the Open Science Framework data repository: https://osf.io/z59us.

The following dataset was generated:

| Author(s) | Year | Dataset title | Dataset URL | Database and Identifier |
|---|---|---|---|---|
| Cook J, Swart J, Froböse M, Diaconescu A, Geurts D, Ouden H, Cools R | 2019 | Catecholaminergic modulation of meta-learning | https://osf.io/z59us | Open Science Framework, z59us |

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

## Appendix 1

### Social and non-social group demographics

**Appendix 1—table 1.** There were no significant differences between the social and non-social groups in age, gender or socioeconomic status as measured using the Barratt Simplified Measure of Social Status (*Barratt, 2012*).

|  | Social | Non-social |  |
| --- | --- | --- | --- |
| *Age in years ($\bar{x}(\sigma)$)* | 21.8 (2.6) | 21.3 (2.0) | t(100) = 1.1, p = 0.3 |
| *Socioeconomic status ($\bar{x}(\sigma)$)* | 60.4 (7.8) | 59.2 (8.4) | t(100) = 0.5, p = 0.6 |
| *N(Male:Female)* | 50 (26:24) | 52 (25:27) | $\chi^2$(1) = 0.16, p = 0.7 |

## Appendix 2

### Instruction scripts

(choice of English or Dutch)

Welcome. You have a choice: either choose the blue shape or the green shape. One shape is correct - guessing which one it is will give you points. When the question mark appears try picking a shape by pressing the left or right button on the response box. [Participant responds]

Feedback: After you make a choice, the correct shape appears in the middle. You didn't really have a lot of information though. When the question mark appears have another go, maybe the same shape will be right again? [Participant responds]

Blue was right again! Things happen in phases in this game. Right now, it looks like you are in a phase where blue is most likely to be correct. Have another go. [Participant responds]

And blue again! It certainly looks as though you are in a blue phase but make sure you pay attention to what the right answers are because the phase that you are in can change at any time. TIP: Here's a tip - ignore which side of the screen the shapes are on - it's the colour that is important! Try again. Perhaps the other shape is right this time? [Participant responds]

Green! This time the green shape was right! The chance of each shape being right or wrong will change as you play, so pay attention! Have another go. ... [Participant responds]

### Social Group

To help you decide - one of the shapes will be highlighted. This is the most popular choice selected by a group of 4 people who previously played this task. Here they think the green shape will be correct. Try picking a shape now [participant responds].

You see? This time the others got it right! Be careful though because we have mixed up the order of the other people's trials so that their choices will also follow phases. Have another go. ... [Participant responds]

It looks like, right now, you could be in a phase where the group's information is pretty useful - perhaps these are trials from the end of their experiment where they had developed a good idea of what was going on. [Participant responds]

Yep it certainly looks like the group information is pretty useful right now but be careful, this could change! Sometimes you will see less useful information - for example from the beginning of their experiment where they didn't have a very good idea of what was going on. [Participant responds]

Getting it right, gives you points. Get enough points and you could earn a silver (extra €2!) or even a gold (extra €4!) prize! Underneath the shapes there is a tracker which shows how close you are to either a gold or silver prize. Every time you get it right the tracker moves a bit closer to the prizes. Pick one more shape and then we'll head to the real game! [Participant responds]

### Non-social Group

To help you decide - one of the shapes will be highlighted. The computer has generated this suggestion using virtual roulette wheels. On each trial the computer spins the roulette, if the ball lands on black the computer will put a frame around the correct answer. If the ball lands on red the computer will frame the incorrect answer. Have another go. .. [Participant responds]

You see? This time the roulette wheel got it right! Be careful though because there are different types of roulette wheel. Some roulette wheels are half red and half black. This type of roulette is equally likely to give you correct and incorrect suggestions. However, others are biased. This type of roulette will give you mostly correct suggestions. [Participant responds]

It looks like, right now, you could be in a phase where the roulette's information is pretty useful - perhaps these are trials from a biased roulette wheel that gives mostly correct information? [Participant responds]

Yep it certainly looks like the roulette's information is pretty useful right now but be careful, this could change! Sometimes you will see less useful information - for example from a roulette wheel that is equally likely to give you correct and incorrect suggestions. [Participant responds]

Getting it right, gives you points. Get enough points and you could earn a silver (extra €2!) or even a gold (extra €4!) prize! Underneath the shapes there is a tracker which shows how close you are to either a gold or silver prize. Every time you get it right the tracker moves a bit closer to the prizes. Pick one more shape and then we'll head to the real game! [Participant responds].

## Appendix 3

## Extended statistical analyses

**Appendix 3—table 1. Win-stay, lose-shift analysis.** Main effects and interactions not involving the factor drug.

| Main effect/ interaction | Statistics | Further information |
|---|---|---|
| Learning type | F (1,100) = 58.595, p < 0.001 | $\beta_{WSLS}$ for experienced ($\bar{x}(\sigma_{\bar{x}})$ = 0.366(0.017)) > inferred ($\bar{x}(\sigma_{\bar{x}})$ = 0.188 (0.012)) |
| Volatility | F (1,100) = 70.671, p < 0.001 | $\beta_{WSLS}$ for volatile ($\bar{x}(\sigma_{\bar{x}})$ = 0.327(0.010)) > stable ($\bar{x}(\sigma_{\bar{x}})$ = 0.227(0.011)) |
| Group | F(1,100) = 4.001, p = 0.048 | $\beta_{WSLS}$ for non-social ($\bar{x}(\sigma_{\bar{x}})$ = 0.295(0.013)) > social ($\bar{x}(\sigma_{\bar{x}})$ = 0.259 (0.013)) |
| Learning type x group | F(1,100) = 7.729, p = 0.006 | $\beta_{WSLS}$ for social ($\bar{x}(\sigma_{\bar{x}})$ = 0.138(0.017)) < non-social ($\bar{x}(\sigma_{\bar{x}})$ = 0.239 (0.017)) for inferred (F(1,100) = 17.743, p < 0.001). No difference between groups for experienced (social $\bar{x}(\sigma_{\bar{x}})$ = 0.381 (0.024), non-social $\bar{x}(\sigma_{\bar{x}})$ = 0.352 (0.024), F(1,100) = 0.694, p = 0.407) |

## Exploration of differences between win-stay and lose-shift behaviour

To explore potential differences between win-stay and lose-shift behaviour we calculated separate scores for win-stay and lose-shift trials. For experienced value learning a trial was denoted as a win-stay if the participant had won on the previous trial and if their response on the current trial was the same (e.g. they selected blue and won on the previous trial, and subsequently re-selected blue). A trial was denoted as a lose-shift if the participant had lost on the previous trial and if they subsequently shifted their response. Win-stay and lose-shift trials were summed and divided by the total number of win/lose trials to control for differing numbers of win/lose trials between randomisation schedules and (volatile/stable) phases. Shapiro-Wilk tests showed that averaged win-stay and lose-shift scores, for experienced-value and inferred-value learning, volatile and stable conditions, and for social and non-social groups, did not significantly deviate from the normal distribution (all p>0.05).

Win-stay and lose-shift scores were entered into a single repeated measures ANOVA with within-subjects factors drug (MPH, PLA), volatility (stable, volatile), learning type (experienced, inferred) and index (win-stay, lose-shift) and between subjects factor group (social, non-social). The interaction between drug, volatility and learning type remained significant (F(1,100) = 7.393, p = 0.008). There was no drug x index interaction (F(1,100) = 0.100, p = 0.470), drug x volatility x index interaction (F(1,100) = 0.096, p = 0.757), or drug x learning type x index interaction (F(1,100) = 1.303, p = 0.256). However, we did observe a main effect of index (F(1,100) = 244.016, p < 0.001) driven by higher win-stay ($\bar{x}(\sigma_{\bar{x}})$ = 0.735 (0.004)) relative to lose-shift scores ($\bar{x}(\sigma_{\bar{x}})$ = 0.567 (0.010)). A main effect of learning type (F (1,100) = 52.705, p < 0.001) driven by higher win-stay and lose-shift scores for experienced-value ($\bar{x}(\sigma_{\bar{x}})$ = 0.691 (0.009)) relative to inferred-value learning ($\bar{x}(\sigma_{\bar{x}})$ = 0.611 (0.006)), and a marginal main effect of group, which approached significance (F(1,100) = 3.742, p = 0.056), driven by higher win-stay and lose-shift scores for the Non-social ($\bar{x}(\sigma_{\bar{x}})$ = 0.661 (0.007)) relative to Social Group ($\bar{x}(\sigma_{\bar{x}})$ = 0.641 (0.007)). These main effects were modulated by a learning type x group interaction (F(1,100) = 6.231, p = 0.014) driven by reduced win-stay and lose-shift scores for the Social relative to Non-social Group for inferred-value learning (social $\bar{x}(\sigma_{\bar{x}})$ = 0.587 (0.009), non-social $\bar{x}(\sigma_{\bar{x}})$ = 0.635 (0.009), F(1,100) = 14.995, p < 0.001), but no difference between groups for experienced-value learning (social $\bar{x}(\sigma_{\bar{x}})$ = 0.695 (0.013), non-social $\bar{x}(\sigma_{\bar{x}})$ = 0.687 (0.012), F(1,100) = 0.177, p = 0.675). A drug x learning type x index x volatility interaction also approached significance (F(1,100) = 3.634, p = 0.059). Post hoc ANOVAS conducted separately for win-stay and lose-shift scores revealed that the

drug x learning type x volatility interaction was primarily driven by effects on lose-shift (drug x learning type x volatility interaction $F_{(1,100)} = 7.589$, $p = 0.007$ (Bonferroni-corrected $p = 0.014$)) rather than win-stay scores ($F_{(1,100)} = 0.073$, $p = 0.787$).

**Appendix 3—table 2.** Learning rate analysis. Main effects and interactions not involving the factor drug. Note that the learning type x volatility interaction is consistent with previous studies of experienced-value learning which have reported higher learning rates in volatile relative to stable environments (*Behrens et al., 2007*; *Behrens et al., 2008*; *Browning et al., 2015*; *Krugel et al., 2009*; *Nassar et al., 2010*).

| Main effect/ interaction | Statistics | Further information |
|---|---|---|
| Learning type | $F_{(1,100)} = 65.409$, $p < 0.001$ | $\alpha_{direct}$ ($\bar{x}(\sigma_{\bar{x}}) = 0.328(0.014)$) > $\alpha_{inferred}$ ($\bar{x}(\sigma_{\bar{x}}) = 0.194(0.010)$) |
| Learning type x volatility | $F_{(1,100)} = 4.882$, $p = 0.029$ | $\alpha_{vol\_experienced}$ ($\bar{x}(\sigma_{\bar{x}}) = 0.341(0.017)$) > $\alpha_{stable\_experienced}$ ($\bar{x}(\sigma_{\bar{x}}) = 0.315(0.015)$; $F_{(1,100)} = 3.916$, $p = 0.051$). No significant difference between $\alpha_{stable\_inferred}$ ($\bar{x}(\sigma_{\bar{x}}) = 0.201(0.012)$) and $\alpha_{vol\_inferred}$ ($\bar{x}(\sigma_{\bar{x}}) = 0.187(0.011)$; $F_{(1,100)} = 1.675$, $p = 0.199$) |
| Learning type x group | $F_{(1,100)} = 4.097$, $p = 0.046$ | $\alpha_{social}$ ($\bar{x}(\sigma_{\bar{x}}) = 0.161(0.014)$) < $\alpha_{non-social}$ ($\bar{x}(\sigma_{\bar{x}}) = 0.227(0.014)$) for inferred ($F_{(1,100)} = 11.349$, $p = 0.001$). No difference between groups for experienced (social $\bar{x}(\sigma_{\bar{x}})0.329(0.021)$, non-social $\bar{x}(\sigma_{\bar{x}}) 0.328(0.020)$; $F_{(1,100)} = 0.002$, $p = 0.966$) |

# Exploration of group differences for participants sensitive to the effect of MPH on inferred-value learning

To guard against the possibility that the lack of a group difference is caused by an insensitivity of inferred-value learning to the effects of MPH, we selected a subgroup of participants who numerically were sensitive to the effects of MPH on inferred-value learning ((MPH-PLA $\alpha_{stable\_inferred}$ + MPH-PLA $\alpha_{volatile\_inferred}$)>(MPH-PLA $\alpha_{stable\_experienced}$+ MPH-PLA $\alpha_{volatile\_experienced}$)). For this subgroup (Social Group N = 26, Non-social Group N = 27) we ran a repeated measures ANOVA with within-subjects factors drug (MPH, PLA), volatility (stable, volatile), and learning type (experienced, inferred) and between-subjects factor group (social, non-social). The interaction between drug, volatility and learning type remained significant ($F_{(1,51)} = 5.506$, $p = 0.023$). There was no drug x volatility x learning type x group interaction ($F_{(1,51)} = 0.813$, $p = 0.371$), no main effect of group ($F_{(1,51)} = 1.415$, $p = 0.240$) and no other significant interactions involving group (all $p > 0.05$; Bayesian t-test comparing groups on scores representing the experienced-inferred difference in the MPH-PLA $\alpha$ difference between volatile and stable $BF_{01} = 2.588$). Under MPH $\alpha_{vol\_experienced}$ ($\bar{x}(\sigma_{\bar{x}}) = 0.306(0.026)$) was greater than $\alpha_{stable\_experienced}$ ($\bar{x}(\sigma_{\bar{x}}) = 0.266(0.020)$), though this difference only approached significance $F_{(1,51)} = 3.474$, $p = 0.068$. Under PLA $\alpha_{vol\_experienced}$ ($\bar{x}(\sigma_{\bar{x}}) = 0.379(0.027)$) and $\alpha_{stable\_experienced}$ ($\bar{x}(\sigma_{\bar{x}}) = 0.397(0.025)$) were comparable ($F_{(1,51)} = 0.714$, $p = 0.402$). There were no differences between $\alpha_{vol\_inferred}$ and $\alpha_{stable\_inferred}$ under MPH ($\alpha_{vol\_inferred}$ $\bar{x}(\sigma_{\bar{x}}) = 0.208(0.019)$, $\alpha_{stable\_inferred}$ $\bar{x}(\sigma_{\bar{x}}) = 0.237(0.019)$, $F_{(1,51)} = 2.559$, $p = 0.116$), or PLA ($\alpha_{vol\_inferred}$ $\bar{x}(\sigma_{\bar{x}}) = 0.170(0.017)$, $\alpha_{stable\_inferred}$ $\bar{x}(\sigma_{\bar{x}}) = 0.166(0.016)$, $F_{(1,51)} = 0.060$, $p = 0.807$).

## Appendix 4

### Model comparison

Data were initially analysed with three different models. Bayesian model selection (BMS), using the VBA toolbox (http://mbb-team.github.io/VBA-toolbox/; *Daunizeau et al., 2014*), was used to evaluate which model provided the best fit to the data. This procedure rests on computing an approximation to the model evidence or $p(y|m)$ the probability of the data $y$ given a model $m$ (*MacKay, 2003*). Here we approximate the log model evidence with F values. The random-effects BMS approach we use here rests on a hierarchical scheme introduced by *Stephan et al. (2009)*, which enables us to estimate the posterior probability that a given model generated the data for any randomly selected participant, relative to all other models considered (for details see *Stephan et al., 2009*) and compute the ''exceedance probability'' that a particular model is more likely than any other model in the comparison set. This exceedance probability represents the amount of evidence that, in the population studied, a given model is more frequent than the others.

- Model 1 comprised a classic Rescorla-Wagner model:
  $$V_{(i+1)} = V_i + \alpha\varepsilon_i$$
- Model 2 extended model 1 with separate learning rates $\alpha$ for volatile and stable blocks.
  $$V_{(i+1)} = V_i + \alpha_{\_volatile}\varepsilon_i + \alpha_{\_stable}\varepsilon_i,$$
  where $\varepsilon_i = (r_i - V_i)$
- Model 3 extended model 2, and had (four) separate learning rates $\alpha$ for stable and volatile blocks for both experienced-value and inferred-value learning:
  $$V_{\_experienced(i+1)} = V_{\_experienced(i)} + \alpha_{\_volatile}\varepsilon_i + \alpha_{\_stable}\varepsilon_i$$
  $$V_{\_inferred(i+1)} = V_{\_inferred(i)} + \alpha_{\_volatile}\varepsilon_i + \alpha_{\_stable}\varepsilon_i$$

Between-conditions random effects Bayesian model comparison (http://mbb-team.github.io/VBA-toolbox/wiki/BMS-for-group-studies/; using function VBA_groupBMC_btwConds.m) was used to test the hypothesis that the same model explained the data produced under both MPH and PLA. For model 3 the exceedance probability was ~1, demonstrating evidence in favour of the conclusion that the data likely originate from the same model (model 3) both under MPH and PLA conditions (*Appendix 4—figure 1*).

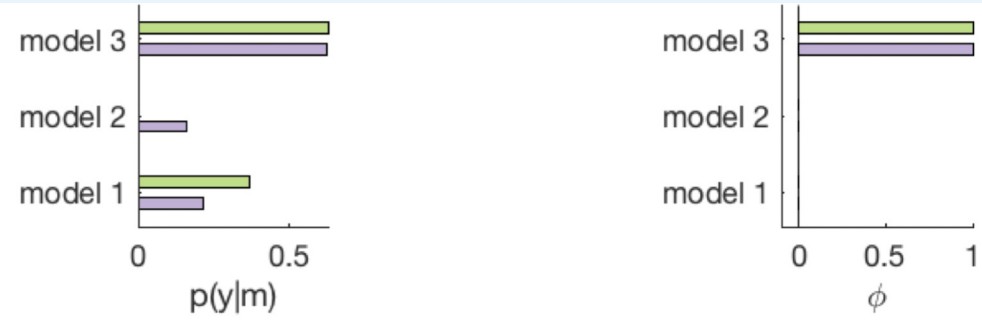

**Appendix 4—figure 1.** $p(y|m)$ = posterior model probability, $\phi$ = exceedance probability, MPH = purple, PLA = green.

## Appendix 5

# The relationship between accuracy and model parameters

To further explore the relationship between accuracy and model parameters we conducted separate regression models for PLA and MPH conditions. PLA accuracy was significantly predicted by a regression model including $\alpha_{stable\_experienced\_PLA}$, $\alpha_{vol\_experienced\_PLA}$, $\alpha_{stable\_inferred\_PLA}$, $\alpha_{volatile\_inferred\_PLA}$, $\beta_{PLA}$ and $\zeta_{PLA}$ as predictors (R = 0.85, F(6,95) = 42.44, p < 0.001; *Appendix 5—table 1*). Within the model the significant predictors were $\alpha_{volatile\_experienced\_PLA}$, $\alpha_{volatile\_inferred\_PLA}$, $\beta_{PLA}$, and $\zeta_{PLA}$ (*Appendix 5—table 1*). Hierarchical regression showed that for $\zeta_{PLA}$ the $R^2$ change value was 0.399 (F change (1,95) = 164.889, p <0.001) indicating that $\zeta_{PLA}$ accounts for an additional 40% of the variance compared to the other regressors in the model. The corresponding value for $\beta_{PLA}$ was 18% ($R^2$change = 0.176, F change (1,95) = 72.741, p <0.001), 2% for $\alpha_{volatile\_experienced\_PLA}$ ($R^2$change = 0.017, F change (1,95) = 7.105, p = 0.009) and 2% for $\alpha_{volatile\_inferred\_PLA}$ ($R^2$change = 0.020, F change (1,95) = 8.063, p = 0.006).

**Appendix 5—table 1.** Coefficients from regression models with MPH and PLA accuracy as dependent variables.

| | $\beta$ | $\beta$(std. Err.) | standardised $\beta$ | t | p |
|---|---|---|---|---|---|
| *PLA* | | | | | |
| *constant* | 0.67 | 0.02 | | 30.15* | <0.001 |
| $\beta_{PLA}$ | −0.25 | 0.02 | −0.66 | −12.84* | <0.001 |
| $\zeta_{PLA}$ | 0.03 | 0.00 | 0.43 | 8.53* | <0.001 |
| $\alpha_{stable\_experienced\_PLA}$ | 0.05 | 0.04 | 0.09 | 1.35 | 0.18 |
| $\alpha_{vol\_experienced\_PLA}$ | 0.10 | 0.04 | 0.18 | 2.67* | 0.01 |
| $\alpha_{stable\_inferred\_PLA}$ | −0.03 | 0.04 | −0.03 | −0.56 | 0.58 |
| $\alpha_{volatile\_inferred\_PLA}$ | 0.13 | 0.05 | 0.16 | 2.84* | 0.01 |
| *MPH* | | | | | |
| *constant* | 0.69 | 0.02 | | 39.88* | <0.001 |
| $\beta_{MPH}$ | −0.26 | 0.02 | −0.78 | −15.92* | <0.001 |
| $\zeta_{MPH}$ | 0.02 | 0.00 | 0.44 | 9.15* | <0.001 |
| $\alpha_{stable\_experienced\_MPH}$ | 0.03 | 0.03 | 0.06 | 1.05 | 0.30 |
| $\alpha_{vol\_experienced\_MPH}$ | 0.10 | 0.03 | 0.22 | 4.05* | <0.001 |
| $\alpha_{stable\_inferred\_MPH}$ | 0.05 | 0.03 | 0.08 | 1.49 | 0.14 |
| $\alpha_{volatile\_inferred\_MPH}$ | 0.03 | 0.04 | 0.05 | 0.92 | 0.36 |

* indicates statistical significance at p < 0.05.

MPH accuracy was significantly predicted by a regression model including $\alpha_{stable\_experienced\_MPH}$, $\alpha_{vol\_experienced\_MPH}$, $\alpha_{stable\_inferred\_MPH}$, $\alpha_{volatile\_inferred\_MPH}$, $\beta_{MPH}$ and $\zeta_{MPH}$ as predictors (R = 0.88, F(6,95) = 55.50, p < 0.001; *Appendix 5—table 1*). Within the model the significant predictors were $\alpha_{volatile\_experienced\_MPH}$, $\beta_{MPH}$, and $\zeta_{MPH}$ (*Appendix 5—table 1*). Hierarchical regression showed that for $\zeta_{MPH}$ the $R^2$change value was 0.551 (F change (1,95) = 253.468, p <0.001) indicating that $\zeta_{MPH}$ accounts for an additional 55% of the variance compared to the other regressors in the model. The corresponding value for $\beta_{MPH}$ was 18% ($R^2$change = 0.182, F change (1,95) = 83.738, p <0.001), and 4% for $\alpha_{volatile\_experienced\_MPH}$ ($R^2$change = 0.036, F change (1,95) = 16.400, p < 0.001).

In sum, regressing model parameters against PLA accuracy demonstrates that under PLA $\zeta$ and, to a lesser extent, $\beta$, $\alpha_{volatile\_experienced}$ and $\alpha_{volatile\_inferred}$ values were predictive of accuracy scores. $\zeta$ and $\beta$ values were also predictive of accuracy scores under MPH but the only learning rate that was predictive of accuracy was $\alpha_{volatile\_experienced}$.

## Appendix 6

### Baseline measures and mood ratings

Details of baseline measures and mood ratings are identical to those reported in *Swart et al. (2017)* and are included here for completeness.

Prior to capsule intake, participants completed a Dutch reading test (NLV, *Schmand et al., 1991*) as a proxy of verbal intelligence on day 1, and the Listening Span Test (*Daneman and Carpenter, 1980*; *Salthouse and Babcock, 1991*), to assess working memory capacity on day 2. Subsequently participants completed a Digit Span Test (forward and backward; *Wechsler, 2008*) and the training phase of a Pavlovian-Instrumental Transfer task (PIT) (*Geurts et al., 2013*; *Huys et al., 2011*) of which data will be reported elsewhere.

Between test days participants completed a number of self-report questionnaires including the Barratt Impulsiveness Scale (BIS-11) (*Patton et al., 1995*) which indexes trait impulsivity. The group that received MPH on day 1 did not differ significantly on any of the baseline measures from the group that received PLA on day 1 (p < 0.05). See *Appendix 6—table 1* for an overview of the neuropsychological test scores and self-report questionnaires.

Mood ratings, heart rate and blood pressure were monitored thrice during each test day, i) before capsule intake, ii) upon start task battery, and iii) upon completion of the task battery. The mood ratings consisted of the Positive and Negative Affect Scale (PANAS) (*Watson et al., 1988*) and the Bond and Lader Visual Analogues Scales (calmness, contentedness, alertness; *Bond and Lader, 1974*), as well as a medical Visual Analogues Scale.

**Appendix 6—table 1.** Mean(standard deviation) scores for neuropsychological tests and self-report questionnaires for the groups that received PLA and MPH on day 1. Significance level for the between group differences are reported. Self-report questionnaires include the Barratt Impulsiveness Scale (BIS-11; *Patton et al., 1995*), the Behavioural Inhibition Scale/Behavioural Activation Scale (BISBAS) (*Carver and White, 1994*), Need for Cognition Scale (NCS) (*Cacioppo et al., 1984*), Multidimensional Scale of Perceived Social Support (MSPSS) (*Zimet et al., 1988*), Barratt Simplified Measure of Social Status (BSMSS) (*Barratt, 2012*), Sociable and Aggressive Dominance Questionnaire (SADQ) (*Kalma et al., 1993*), Beck Depression Inventory II (BDI-II) (*Beck et al., 1996*), Spielberger Trait Anxiety Inventory (STAI) (*Spielberger et al., 1983*).

|  |  | Group 1 PLA Day 1 | Group 2 MPH Day 1 |  |
| --- | --- | --- | --- | --- |
| *Neuropsychological tests* | Listening span | 5.0 (0.9) | 4.6 (1.2) | p = 0.160 |
|  | NLV | 94.4 (7.6) | 92.6 (7.6) | p = 0.230 |
|  | Digit span – forward | 17.2 (3.7) | 16.2 (3.6) | p = 0.158 |
|  | Digit Span - backward | 14.7 (3.4) | 13.9 (2.7) | p = 0.219 |
| *Self-report questionnaires* | Impulsivity (BIS-11) | 63.5 (8.9) | 60.2 (7.9) | p = 0.052* |
|  | Behavioural inhibition (BIS) | 16.4 (3.7) | 16.3 (3.5) | p = 0.899 |
|  | Behavioural activation (BAS) | 22.8 (3.9) | 23.9 (4.0) | p = 0.166 |
|  | Need for cognition (NCS) | 64.5 (10.5) | 62.2 (10.5) | p = 0.264 |
|  | Social support (MDSPSS) | 71.1 (10.1) | 69.3 (9.6) | p = 0.349 |
|  | Social status (BSMSS) | 49.8 (12.1) | 45.9 (12.7) | p = 0.114 |
|  | Social dominance (SADQ) | 4.1 (0.9) | 4.1 (0.8) | p = 0.823 |
|  | Aggressive dominance (SADQ) | 2.6 (0.6) | 2.6 (0.6) | p = 0.691 |
|  | Depressive symptoms (BDI) | 3.5 (3.7) | 3.6 (3.9) | p = 0.969 |
|  | Anxiety symptoms (STAI) | 32.4 (6.6) | 32.4 (7.2) | p = 0.995 |

*One participant had an outlying score on the BIS-11. Without outlier: p = 0.090.

## Drug effects on mood and medical symptoms

For this control analysis we performed a repeated measures MANOVA using Pillai's trace with the within-subject factors Time (baseline/start testing/end testing) and Drug (MPH/placebo), and dependent variables Positive Affect, Negative Affect, Calmness, Contentedness, Alertness, and Medical Symptoms. Significant effects were further explored with Bonferonni corrected repeated measures ANOVA, where alpha = 0.05/6 $\approx$ .008. Greenhouse-Geisser correction was applied when the assumption of sphericity was not met.

MPH affected self-report ratings ($V = 0.38$, $F(12,90) = 4.7$, $p < 0.001$), in the absence of baseline differences ($V = 0.07$, $F(6,96) = 1.1$, $p = 0.359$). After capsule intake MPH increased Positive Affect ($F(1,101) = 17.5$, $p < 0.001$), Alertness ($F(1,101) = 15.2$, $p < 0.001$), and Medical Symptoms ($F(1,101) = 11.1$, $p = 0.001$), and decreased Calmness ($F(1,101) = 8.6$, $p = 0.004$), relative to PLA.

## Drug effects as a function of impulsivity and working memory capacity

Effects of drugs, like MPH, which act on the dopamine system have previously been reported to vary as a function of baseline dopamine levels (*Cools and D'Esposito, 2011*; *van der Schaaf et al., 2013*). To assess these potential sources of individual variability in MPH effects, we took into account two measures that have been demonstrated with positron emission tomography to relate to dopamine baseline function: working memory capacity for its relation to striatal dopamine synthesis capacity (*Cools et al., 2008*; *Landau et al., 2009*) and trait impulsivity for its relation to dopamine (auto)receptor availability (*Buckholtz et al., 2010*; *Kim et al., 2014*; *Reeves et al., 2012*). Working memory capacity was assessed using the Listening Span Test (*Daneman and Carpenter, 1980*; *Salthouse and Babcock, 1991*) and impulsivity was assessed using the BIS-11 (*Patton et al., 1995*) as detailed above.

Z-scored BIS-11 and Z-scored listening span were entered as covariates into two separate ANCOVAs with $\beta_{WSLS}$ as dependent variable, within-subjects factors drug (MPH, PLA), volatility (stable, volatile), and learning type (experience, inferred) and between-subjects factor group (social, non-social). This analysis showed no main effects of, or interactions involving, BIS-11 (all p > 0.05). There was, however, a marginally significant interaction between listening span scores and drug ($F(1,99) = 3.912$, $p = 0.051$). This was driven by a significant positive relationship between listening span and $\beta_{WSLS\_MPH-PLA}$ (Pearson's r = 0.225, p = 0.023). In other words, participants with a high listening span (putatively high DA synthesis capacity) showed the greatest increases in WSLS strategy under MPH compared to PLA. More specifically, there was a negative relationship between listening span and $\beta_{WSLS\_PLA}$ (r = -0.273, p = 0.005), which was ameliorated under MPH (no correlation between listening span and $\beta_{WSLS\_MPH}$; r = -0.03, p = 0.765). There were no other significant interactions involving listening span and no main effect of listening span (all p>0.05).

## Dominance and social learning

In previous work (*Cook et al., 2014*) we reported a positive relationship between social learning and social dominance (SD), and a negative relationship between social learning and aggressive dominance (AD) as indexed by the Sociable and Aggressive Dominance Questionnaire (SADQ; *Kalma et al., 1993*). To create an analogous analysis we focused solely on the PLA condition for the Social Group. Social learning was indexed by regressing a Bayesian Learner Model (*Behrens et al., 2007*) of optimal responses based on the social information (the red frame) alone against each participant's responses. SD and AD we measured using the SADQ. Although we replicated the negative relationship between (z-scored) AD and social learning (r = −0.30, p = 0.034), there was no significant correlation between (z-scored) SD and social learning (r = −0.14, p = 0.339). However, it should be noted that the current study used an un-validated Dutch translation of the SADQ. In contrast with our previous work (*Cook et al., 2014*) and work of others (*Kalma et al., 1993*), a

correlation between SD and AD was observed in the current sample (r = 0.346, p < 0.001). Further work is necessary to understand whether the non-replication of the significant SD and social learning relationship is due to the use of the un-validated translation.

