## [Decision Letter]

**Acceptance summary:**

Optimal learning rate during probabilistic reversal depends on the volatility of the environment, namely, the frequency of reversal. Therefore, for the best performance, the learning rate should be adjusted according to the volatility. In this paper, Cook et al., tested the hypothesis that catecholamine level influences the ability to adjust the learning rate according to volatility by testing the effect of MPH on normal subjects. The results described in this manuscript support the hypothesis.

**Decision letter after peer review:**

Thank you for submitting your article "Catecholaminergic modulation of meta-learning" for consideration by *eLife*. Your article has been reviewed by three peer reviewers, including Daeyeol Lee as the Reviewing Editor and Reviewer #1, and the evaluation has been overseen by a Reviewing Editor and Kate Wassum as the Senior Editor. The following individual involved in review of your submission has agreed to reveal their identity: Peter Murphy (Reviewer #2).

The reviewers have discussed the reviews with one another and the Reviewing Editor has drafted this decision to help you prepare a revised submission.

Summary:

Optimal learning rate during probabilistic reversal depends on the volatility of the environment, namely, the frequency of reversal. Therefore, for the best performance, the learning rate should be adjusted according to the volatility. In this paper, Cook et al., tested the hypothesis that catecholamine level influences the ability to adjust the learning rate according to volatility by testing the effect of MPH on normal subjects. Overall, the results are largely consistent with the hypothesis. The large sample size and highly rich experimental design are especially laudable aspects of this study.

Essential revisions:

1) While the authors discuss the relationship between reinforcement learning and working memory, it remains unclear whether the effect of MPH demonstrated in this study is primarily driven its effect on working memory or meta-learning. One possible strategy to examine this question might be to use a logistic regression model to examine the delayed (or lagged) WSLS strategy, for example, by including the effect of WSLS from the outcome in trial t-2, t-3, etc. The effect on learning rate should be reflected not only in the coefficient for the preceding (t-1) trial, but for earlier trials as well.

2) Similarly, recent results from both animal and human studies have demonstrated the learning rate as well as WSLS might depend on the valence of outcomes (i.e., win vs. loss). Therefore, it would be useful for the authors to test the effect of MPH on win-stay vs. lose-switch as well as the model with differential learning rates for rewarded and unrewarded trials. A recent rodent study has even identified specific neural circuits that might underlie learning from positive and negative outcomes (Groman et al., 2019).

3) It is alluded to at various points that the main conclusion to be drawn here is that catecholamines help to "optimize" learning rate as a function of volatility. Can the authors provide the values of learning rates optimal for stable and volatile conditions in the task used in this study? What are the win-stay/lose-switch betas and learning rates (both 'experienced' and 'inferred') for the ideal observer model for this task, estimated in an identical way as for the human participants? Do the participants come close to the ideal observer in placebo, or do they exhibit systematic biases in their behavior that the drug counteracts? Are they weighting the different sources of information (direct vs indirect) appropriately, or do they afford one more weight than they 'should'? Addressing these questions by explicitly comparing human behavior to the ideal observer model would in my opinion make for a more complete manuscript and make claims about the drug helping participants to optimize behavior much easier to evaluate.

4) The authors provided interesting information as to potential factors related to the effect of MPH on accuracy, but it would be better if there are a bit more information about the effect of various model parameters on accuracy and how these parameters were affected by MPH. For example, were the inverse temperature (β) and strategy-weight parameter (zeta) affected by MPH? How strongly were these parameters in either MPH or PLA group related with accuracy? Without this information, it remains difficult to appreciate the last sentence in the subsection “Summary of results” ("… the contribution of learning rate to accuracy is minimal.").

5) The main effect of MPH is to selectively decrease learning rates in stable contexts, and not to increase learning rates (either complementarily in volatile contexts if the meta-learning account is to be believed, or across the board if catecholamines play a more restricted role one level lower in the hierarchy – i.e. unidirectionally shifting the learning rate). This key aspect of the results warrants more extensive discussion. Perhaps the ideal observer analysis suggested above might shed some light. If under placebo participants already well-approximate the ideal observer in volatile contexts but over-estimate volatility in stable contexts, this perhaps lends additional credence to the meta-learning account – i.e. that the drug helps participants to learn the appropriate level of volatility in the context in which they 'need help'.

6) The description of the 'model-free' win-stay/lose-switch analysis is somewhat vague. Are betas for each learning type calculated from the same multiple regression model, or different univariate regressions? The former would seem more appropriate, since learning type constitutes two components (direct vs indirect) that in principle should contribute to the same choice. Authors' rationale for introducing this model in the way they do currently – i.e. to differentiate participants whose choices were primarily driven by experienced vs. inferred value – also needs to be explained more clearly. Isn't exactly this differentiation provided by the epsilon parameter in the existing δ-rule model fits? How are the results from these different models related?

7) The authors report that they measured heart rate before drug administration, and both before and after the task battery. These data are relevant for evaluating both the efficacy of the drug manipulation and the effects reported on fitted model parameters. Did participants whose heart rate was most significantly affected by the drug also show the strongest behavioral effects, or perhaps additional behavioral effects not apparent at the full-sample level? Taking such data into account could greatly improve the sensitivity of the reported analyses.

8) It is striking, but not emphasized, that the authors do not replicate the original study under placebo: there is no effect of volatility on the learning rate for experienced feedback under placebo (Figure 3A green, Results section). It seems important to emphasize this lack of replication and discuss it.

9) The authors should not overstate the adaptiveness of increasing learning rate or changing strategy between stable and volatile environments (e.g. subsection “Win-stay, lose-shift analysis”, and in a few other places). As the authors show at the end, the learning rate effects don't actually seem to make an important difference in this task, so "optimality" claims should be weakened.

10) It would be useful to present more general results before jumping into more specific analyses (win-stay-lose-shift). It might be better to present the information about the overall performance in the task before describing drug effects on performance. Some of the figures in the supplement, including model validation, could usefully be moved to main text.

---

## [Author Response]

Essential revisions:1) While the authors discuss the relationship between reinforcement learning and working memory, it remains unclear whether the effect of MPH demonstrated in this study is primarily driven its effect on working memory or meta-learning. One possible strategy to examine this question might be to use a logistic regression model to examine the delayed (or lagged) WSLS strategy, for example, by including the effect of WSLS from the outcome in trial t-2, t-3, etc. The effect on learning rate should be reflected not only in the coefficient for the preceding (t-1) trial, but for earlier trials as well.

We thank the reviewers for this suggestion. The study was not specifically set up to disentangle effects of MPH on reinforcement learning versus working-memory (model-) based learning. Thus, as the reviewer notes, whilst we have discussed both, we have avoided trying to pin our effects to one or the other. The reviewer’s suggestion, however, may shed some light on the issue. To examine the lagged WSLS strategy, for each participant, three regressors were created for each drug condition (MPH, PLA) and volatility level (volatile, stable). Regressor 1 represented the choice that the participant should have made, for each trial, if they switched after losing on the previous trial (t-1) and stayed after winning on t-1. Regressor 2 represented the choices that the participant should have made if they switched after losing on trial t-2 and stayed after winning on t-2. Regressor 3 represented the choices that the participant should have made if they switched after losing on trial t-3 and stayed after winning on t-3. For each participant we therefore calculated three β_WSLS t-n_ values where a high value means that the corresponding regressor explains a large amount of variance in the participant’s choices.

To evaluate whether the interaction between drug and volatility on WSLS values for experienced-value learning differs across lag (t-1, t-2, t-3) we ran a repeated measures ANOVA with experienced value β_WSLS t-n_ as the dependent variable and within-subjects factors volatility (volatile, stable), drug (MPH, PLA) and lag (t-1, t-2, t-3). If the interaction between drug and volatility were specific to t-1 we would have observed a significant drug, volatility and lag interaction such that the drug x volatility interaction is present only at t-1. We observed no such significant interaction (F(2,202) = 1.05, p = 0.353; BF01 = 15.09).

2) Similarly, recent results from both animal and human studies have demonstrated the learning rate as well as WSLS might depend on the valence of outcomes (i.e., win vs. loss). Therefore, it would be useful for the authors to test the effect of MPH on win-stay vs. lose-switch as well as the model with differential learning rates for rewarded and unrewarded trials. A recent rodent study has even identified specific neural circuits that might underlie learning from positive and negative outcomes (Groman et al., 2019).

We thank the reviewer for this interesting suggestion. Indeed, in Appendix 3 we reported the win-stay, lose-shift analysis in which we included measure (WS, LS) as a factor. Here we observed a drug x learning type x index x volatility interaction which approached statistical significance (F(1,100) = 3.634, p = 0.059). Post hoc ANOVAS conducted separately for win-stay and lose-shift scores revealed that the drug x learning type x volatility interaction was significant for lose-shift but not win-stay scores. However, we advise caution with respect to this analysis because our paradigm is underpowered for detecting differences between rewarded and unrewarded trials. The abovementioned WSLS analysis has 8 parameters (WS__vol_experienced_, WS__stable_experienced_, LS__vol_experienced_, LS__stable_experienced_, WS__vol_inferred_, WS__stable_inferred_, LS__vol_inferred_, LS__stable_inferred_), with only 120 trials, that means 15 trials per cell of the design. A learning model with separate learning rates for both volatile and stable phases and rewarded and unrewarded trials would have 10 free parameters (8 learning rates, plus βand ) and thus 12 trials per parameter. Indeed, Bayesian model selection (Daunizeau, Adam and Rigoux, 2014) – where log model evidence is approximated with F values – demonstrates that our existing adapted Rescorla Wagner model (with separate learning rates for volatile and stable) provides a better fit to the data than a model which has separate learning rates for volatile/stable and rewarded/unrewarded trials (see Author response image 1 for probability of the data given the model (p(y∣m) and exceedance probability (*ϕ*)). To avoid making potentially unreliable claims we chose not to include such an analysis in the main text of our manuscript.

3) It is alluded to at various points that the main conclusion to be drawn here is that catecholamines help to "optimize" learning rate as a function of volatility. Can the authors provide the values of learning rates optimal for stable and volatile conditions in the task used in this study? What are the win-stay/lose-switch betas and learning rates (both 'experienced' and 'inferred') for the ideal observer model for this task, estimated in an identical way as for the human participants? Do the participants come close to the ideal observer in placebo, or do they exhibit systematic biases in their behavior that the drug counteracts? Are they weighting the different sources of information (direct vs indirect) appropriately, or do they afford one more weight than they 'should'? Addressing these questions by explicitly comparing human behavior to the ideal observer model would in my opinion make for a more complete manuscript and make claims about the drug helping participants to optimize behavior much easier to evaluate.

By fitting our model to synthetic data which followed the probabilistic schedules we calculated optimal learning rates for the adapted Rescorla-Wagner model. The average (over 100 synthetic datasets) optimal learning rates were α_optimal_experienced_stable_ = 0.16, α_optimal_ experienced_volatile_ = 0.21, α_optimal_inferred_stable_ = 0.17, α_optimal_inferred_volatile_ = 0.19. __optimal_ (the weight of inferred- relative to experienced-value learning) and β_optimal_ were 0.23 and 1.44 respectively. For β_WSLS_ the optimal values wereβ_WSLS_experienced_stable_ = 0.01, β_WSLS_experienced_volatile_ = 0.32, β_WSLS_inferred_stable_ = -0.01 and β_WSLS_inferred_volatile_ = 0.30.

By calculating the difference between learning rates and the corresponding optimal αwe created indices of the ‘distance from optimal’ for each drug condition, volatility level and learning type and submitted these to a 2 (drug) x 2 (volatility) x 2 (learning type) x 2 (group) ANOVA. We observed a significant drug x learning type x volatility interaction (F(1,100) = 6.913, p = 0.010). For experienced value learning there was a significant drug x volatility interaction (F(1,100) = 6.151, p = 0.015). This interaction was not significant for inferred-value learning (F(1,100) = 1.588, p = 0.211). Simple effects analyses demonstrated that, experienced-value learning rates under PLA were further from the optimal value in the stable compared to volatile phase (α_volatile_experienced-optimal_ x-(σx-) = 0.119(0.019), α_stable_experienced-optimal_ x-(σx-) = 0.171(0.019), F(1,100) = 11.227, p = 0.001). The difference between volatile and stable was not significant under MPH (α_volatile_experienced-optimal_ x-(σx-) = 0.143(0.020), α_stable_experienced-optimal_ x-(σx-) = 0.139(0.017), F(1,100) = 0.051, p = 0.823). Numerically, stable learning rates were closer to optimal under MPH relative to PLA, however, this effect only approached significance (F(1,100) = 3.218, p = 0.076). The distance from optimal for volatile learning rates was not significantly affected by MPH (F(1,100) = 1.387, p = 0.242).

Under PLA, and β values were significantly greater than the optimal values (__PLA-optimal_ x- = 0.282(0.027), t(101) = 10.434, p < 0.001; β__PLA-optimal_ x- = 1.648, t(101) = 9.873, p < 0.001). Paired t-tests showed no significant effect of MPH for either (t(101) = 0.554, p = 0.581), or β values (t(101) = 1.152, p = 0.252).

To summarise, in line with the reviewer’s intuition, there was a trend such that under PLA participants systematically over-estimated the volatility of the stable environment and MPH counteracted this systematic bias.

We thank the reviewer for suggesting this analysis which we think has brought much clarity to the manuscript. We have now added the following text and Figure 4:

Subsection “Optimal Learner Analysis”

“In the above section (Learning rate analysis) we report that the difference inα_volatile_experienced_ and α_stable_experienced_ is greater under MPH compared to PLA. To investigate whether, in this particular task, a greater difference between α_volatile_experienced_ and α_stable_experienced_ can be considered ‘more optimal’ we created an optimal learner model with the same architecture and priors as the adapted Rescorla-Wagner model employed above. By generating 100 synthetic datasets, which followed our probabilistic schedules, and fitting the optimal learner model to the synthetic data, we estimated that the average optimal learning rates were α_optimal_experienced_stable_ = 0.16, α_optimal_ experienced_volatile_ = 0.21, α_optimal_inferred_stable_ = 0.17, α_optimal_inferred_volatile_ = 0.19. Optimal values for α_optimal_ (the weight of inferred- relative to experienced-value learning) and β_optimal_ were 0.23 and 1.44 respectively.

‘Distance from optimal’ indices (the difference between α and α_optimal_) were submitted to a 2 (drug: MPH, PLA) x 2 (volatility: volatile, stable) x 2 (learning type: experienced, inferred) x 2 (group: social, non-social) ANOVA. We observed a significant drug x learning type x volatility interaction (F(1,100) = 6.913, p = 0.010; Figure 4). Only for experienced-value learning was there a significant drug x volatility interaction (F(1,100) = 6.151, p = 0.015; inferred-value learning F(1,100) = 1.588, p = 0.211). Simple effects analyses demonstrated that, experienced-value learning rates under PLA were further from the optimal value in the stable compared to volatile phase (α_volatile_experienced-optimal_ x-(σx-) = 0.119(0.019), α_stable_experienced-optimal_ x-(σx-) = 0.171(0.019), F(1,100) = 11.227, p = 0.001). The difference between volatile and stable was not significant under MPH (α_volatile_experienced-optimal_ x(σ) = 0.143(0.020), α_stable_experienced-optimal_ x-(σx-) = 0.139(0.017), F(1,100) = 0.051, p = 0.823). Numerically, stable learning rates were closer to optimal under MPH relative to PLA, however, this effect only approached statistical significance (F(1,100) = 3.218, p = 0.076). The distance from optimal for volatile learning rates was not significantly affected by MPH (F(1,100) = 1.387, p = 0.242). In sum, there was a trend such that under PLA participants systematically over-estimated the volatility of the stable environment and MPH counteracted this systematic bias.”

4) The authors provided interesting information as to potential factors related to the effect of MPH on accuracy, but it would be better if there are a bit more information about the effect of various model parameters on accuracy and how these parameters were affected by MPH. For example, were the inverse temperature (β) and strategy-weight parameter (zeta) affected by MPH? How strongly were these parameters in either MPH or PLA group related with accuracy? Without this information, it remains difficult to appreciate the last sentence in the subsection “Summary of results” ("… the contribution of learning rate to accuracy is minimal.").

We have now included further analyses in which we show the strength of the relationship between model parameters and accuracy under MPH and PLA separately (previously we only showed this data in terms of MPH-PLA differences). Consequently, we have added the following text to the manuscript as Appendix 5.

“To further explore the relationship between accuracy and model parameters we conducted separate regression models for PLA and MPH conditions. PLA accuracy was significantly predicted by a regression model including α_stable_experienced_PLA,_ α_vol_experienced_PLA,_ α_stable_inferred_PLA,_ α_volatile_inferred_PLA,_ β_PLA_ andζ_PLA_ as predictors (R = 0.85, F(6,95) = 42.44, p < 0.001; Appendix 5 – Table 1.). Within the model the significant predictors wereα_volatile_experienced_PLA,_α_volatile_inferred_PLA_,β_PLA_, andζ_PLA_ (Appendix 5 – Table 1.). Hierarchical regression showed that for ζ_PLA_ the R^2^change value was 0.399 (F change (1,95) = 164.889, p <0.001) indicating that ζ_PLA_ accounts for an additional 40% of the variance compared to the other regressors in the model. The corresponding value for β_PLA_ was 18% (R^2^change = 0.176, F change (1,95) = 72.741, p <0.001), 2% for α_volatile_experienced_PLA_ (R^2^change = 0.017, F change (1,95) = 7.105, p = 0.009) and 2% for α_volatile_inferred_PLA_ (R^2^change = 0.020, F change (1,95) = 8.063, p = 0.006).

MPH accuracy was significantly predicted by a regression model including α_stable_experienced_MPH,_ α_vol_experienced_MPH,_ α_stable_inferred_MPH,_ α_volatile_inferred_MPH,_ β_MPH_ andζ_MPH_ as predictors (R = 0.88, F(6,95) = 55.50, p < 0.001; Appendix 5 – Table 1.). Within the model the significant predictors were α_volatile_experienced_MPH_,β_MPH_, andζ_MPH_ (Appendix 5 – Table 1.). Hierarchical regression showed that for ζ_MPH_ the R^2^change value was 0.551 (F change (1,95) = 253.468, p <0.001) indicating that ζ_MPH_ accounts for an additional 55% of the variance compared to the other regressors in the model. The corresponding value for β_MPH_ was 18% (R^2^change = 0.182, F change (1,95) = 83.738, p <0.001), and 4% for α_volatile_experienced_MPH_ (R^2^change = 0.036, F change (1,95) = 16.400, p < 0.001).

In sum, regressing model parameters against PLA accuracy demonstrates that under PLA ζ and, to a lesser extent, β, α_volatile_experienced_ and α_volatile_inferred_ values were predictive of accuracy scores. ζ and β values were also predictive of accuracy scores under MPH but the only learning rate that was predictive of accuracy was α_volatile_experienced_.”

In the main text (subsection “Relationship between accuracy scores and model-based analyses”) we have edited the following paragraph to improve clarity:

“Regressing model parameters against MPH-PLA accuracy demonstrates that any differences in accuracy scores between MPH and PLA conditions were predominately driven by effects of MPH on ζ (38%) and, to a lesser extent, β (13%) and α_volatile_experienced_ (7%). In other words, participants who showed the greatest benefit of the drug in terms of accuracy scores were those who experienced the greatest decrease in ζ under MPH (where a decrease corresponds to a greater bias towards the use of experienced, as opposed to inferred, learned values). As noted above, there was no significant effect of drug on ζ across all participants (ζ_MPH_(x-(σx-) = 0.499(0.027), ζ_PLA_x-(σx-) = 0.512(0.027); t(101) = -0.554, p = 0.581). The finding that changes in accuracy are accounted for by only a very small proportion of the variance in learning rate is consistent with the observation that the drug altered learning rate without altering accuracy.”

In addition, we have clarified the sentence that the reviewer refers to so that it now reads …

“Although the drug significantly affected learning rate adaptation it did not affect accuracy, potentially because, for the current task, accuracy is predominantly influenced by the weighting of experience-value relative to inferred-value learning (i.e. ζ values).” (subsection “Summary of results”).

5) The main effect of MPH is to selectively decrease learning rates in stable contexts, and not to increase learning rates (either complementarily in volatile contexts if the meta-learning account is to be believed, or across the board if catecholamines play a more restricted role one level lower in the hierarchy – i.e. unidirectionally shifting the learning rate). This key aspect of the results warrants more extensive discussion. Perhaps the ideal observer analysis suggested above might shed some light. If under placebo participants already well-approximate the ideal observer in volatile contexts but over-estimate volatility in stable contexts, this perhaps lends additional credence to the meta-learning account – i.e. that the drug helps participants to learn the appropriate level of volatility in the context in which they 'need help'.

We are extremely grateful to the reviewer for suggesting the optimal learner analysis which we think has brought a considerable amount of clarity to the results. The reviewer’s suspicion, that participants were particularly liable to over-estimate volatility in the stable context was confirmed with our analysis. We have now added the analysis and figure detailed above (Query 3) to the Results section and the following text to the Discussion section:

“Interestingly, we observed that under PLA, participants exhibited a bias towards over-estimating volatility (learning rates were generally greater than the corresponding optimal learner estimates). This bias was significantly greater for stable relative to volatile phases. MPH reduced learning rates in the stable context thereby bringing the stable learning rate closer to the optimal value. MPH did not bring learning rates closer to the optimal values in the volatile context. It is thus possible that, in line with a meta-learning account, MPH optimises learning rate but that, across the group, this effect could only be observed in the stable context where learning rates were especially non-optimal.” (subsection “MPH enhances learning rate adaptation as a function of environmental volatility”).

6) The description of the 'model-free' win-stay/lose-switch analysis is somewhat vague. Are betas for each learning type calculated from the same multiple regression model, or different univariate regressions? The former would seem more appropriate, since learning type constitutes two components (direct vs indirect) that in principle should contribute to the same choice. Authors' rationale for introducing this model in the way they do currently – i.e. to differentiate participants whose choices were primarily driven by experienced vs. inferred value – also needs to be explained more clearly. Isn't exactly this differentiation provided by the epsilon parameter in the existing δ-rule model fits? How are the results from these different models related?

We have now clarified the description of the win-stay, lose-shift analysis as follows:

“For each participant, for each of the 8 cells of the 2x2x2 factorial design (drug x volatility x learning type), a regressor was created representing the choice that they would have made, for each trial, if they were using a WSLS strategy. For experience-value learning, if on the previous trial the participant had chosen blue and won, the regressor would predict that the participant should stay with a blue choice (coded as 1); if, however, the participant had chosen blue and lost, the regressor would predict a shift to a green choice (coded as 0). Similarly, for inferred-value learning, if the participant had previously followed the frame’s advice (coded as 1) and lost the regressor would predict that the participant should shift such that they do not follow the frame’s advice (coded as 0) in the current trial. Regressing these predictors against participant’s actual choices, in a series of 8 separate univariate regressions, provided 8 β values (β_WSLS_) per participant where high values indicate that a WSLS strategy accounts for a large amount of variance in the participant’s choices.” (subsection “Win-stay, lose-shift analysis”)

Should readers wish to re-create this analysis themselves the scripts will be accessible via the Open Science Framework upon publication of this manuscript. In addition, we have clarified our rationale for this analysis and its relationship with the adapted Rescorla-Wagner model as follows:

“A participant who unerringly follows a WSLS strategy is completely guided by their performance on the previous trial: If they choose blue and win they stay with a blue choice; if they choose blue and lose, they shift to a green choice. In a classic reinforcement learning model (e.g. Rescorla and Wagner, 1972) such a participant would exhibit a learning rate equal to 1. That is, they would be completely ‘forgetting’ about past events and reacting only to trial t-1. The extent to which participants employ a WSLS strategy typically correlates with their learning rate, thus WSLS can be (cautiously) used as a proxy for learning rate. Our first analysis therefore investigated the influence of catecholaminergic modulation of meta-learning by quantifying the effect of MPH on the use of this simple learning strategy in volatile and stable environments.”

To ensure that different analysis structures (i.e. single multiple regression versus multiple univariate regressions) do not change the pattern of results we have re-run our analysis as a single multiple regression model with blue/green response as the dependent variable. Specifically, 7 regressors were created representing, for each trial (240 trials spanning days one and two 1) drug, 2) WSLS_experienced_ x volatility_experienced_ 3) WSLS_experienced_ x drug, 4) WSLS_experienced_ x volatility_experienced_ x drug, 5) WSLS_inferred_ x volatility _inferred_ 6) WSLS_inferred_ x drug, 7) WSLS_inferred_ x volatility_inferred_ x drug. For each regressor we ran a repeated measures ANOVA with within-subjects factor learning type (experience-value, inferred-value) and between-subjects factor group (social, non-social).

Importantly, this analysis recapitulated the drug x volatility x learning type interaction that we have reported in the main manuscript (subsection “Win-stay, lose-shift analysis*”*). That is, there was a main effect of learning type for the WSLS x volatility x drug regressor (F(1,99) = 5.749, p = 0.018). Post-hoc one-sample t-tests demonstrated that for experienced-value learning the βvalue was significantly greater than zero (x-σx- = 0.041(0.017), t(100) = 2.406, p = 0.018). βfor inferred-value learning did not differ from zero (x-(σx-) = -0.078(0.049), t(100) = -1.602, p = 0.112). This result parallels our main analysis in which we show an interaction between drug and volatility for experienced-value, but not inferred-value learning. Further, as in our main analysis, we did not observe an interaction between learning type and group (F(1,99) = 0.032, p = 0.858), or a main effect of group (F(1,99) = 0.069, p = 0.793).

7) The authors report that they measured heart rate before drug administration, and both before and after the task battery. These data are relevant for evaluating both the efficacy of the drug manipulation and the effects reported on fitted model parameters. Did participants whose heart rate was most significantly affected by the drug also show the strongest behavioral effects, or perhaps additional behavioral effects not apparent at the full-sample level? Taking such data into account could greatly improve the sensitivity of the reported analyses.

The heartrate measures that we acquired served the purpose of altering us to any adverse effects of the drug. Correspondingly, we checked that participants’ vital signs were within the following ranges:

Syst. BP 95-180 mm HgDiast. BP 50-95 mm HgHeart rate 45-120 bpm

If this was the case, we checked a box on our monitoring form, if not we called for medical assistance. Consequently, we do not have this data in a format suitable for further analysis. Nevertheless, we think the reviewers’ suggestion, to explore whether our main results vary according to individual differences in drug efficacy, is a good one. To address this, we have used positive affect ratings as an index of drug efficacy. Across all participants we observed that, after capsule intake, MPH increased Positive Affect (*F*(1,101) = 17.5, *p* <.001) relative to PLA (Appendix 6). To index individual differences in drug responsivity we calculated, for each participant, the difference between MPH and PLA conditions in this change in positive affect and included it as a covariate in our main ANOVA with learning rate as dependent variable. We observed no main effect of this covariate (*F*(1,98) = 0.598, *p* = 0.441), and no significant interactions involving the covariate (all p > 0.135). With this covariate included in the analysis our main interaction of interest (drug x volatility x learning type) remained statistically significant (*F*(1,98) = 7.182, *p* = 0.009).

8) It is striking, but not emphasized, that the authors do not replicate the original study under placebo: there is no effect of volatility on the learning rate for experienced feedback under placebo (Figure 3A green, Results section). It seems important to emphasize this lack of replication and discuss it.

Many thanks to the reviewers for highlighting this important point. We have now added the following text to the Discussion section:

“Our optimal learner analysis also demonstrated that for optimal performance α_volatile_ should be greater than α_stable._ Consequently, it is striking that under PLA our participants did not exhibit a significant difference between α_volatile_ and α_stable_. By fitting learning models to participants’ responses, previous studies have demonstrated higher learning rates in volatile compared to stable phases (e.g. Browning et al., 2015; Diaconescu et al., 2014). However, such studies, differ from ours in that demands for learning were lower than those in the current study: Participants learned from only one source of information (Browning et al., 2015) or were provided with explicit outcome probabilities (Diaconescu et al., 2014). To the best of our knowledge, the current experiment is the first to estimate learning rates, for volatile and stable phases, when participants are simultaneously learning from two sources of information. Here we demonstrated a lack of adaptation to the volatility of the environment under PLA, (possibly due to high demands for learning from dual sources), which is rescued by MPH.”

9) The authors should not overstate the adaptiveness of increasing learning rate or changing strategy between stable and volatile environments (e.g. subsection “Win-stay, lose-shift analysis”, and in a few other places). As the authors show at the end, the learning rate effects don't actually seem to make an important difference in this task, so "optimality" claims should be weakened.

Throughout the manuscript we have replaced the term ‘learning rate optimisation’ with ‘learning rate adjustment’. The latter captures changes in learning rate without implying that these changes result in more optimal performance. In addition, we have added the optimal learner analysis and text relating to this in the Discussion section (see Queries 3 and 5 above).

10) It would be useful to present more general results before jumping into more specific analyses (win-stay-lose-shift). It might be better to present the information about the overall performance in the task before describing drug effects on performance. Some of the figures in the supplement, including model validation, could usefully be moved to main text.

Many thanks for highlighting this. We have now restructured the manuscript such that results relating to overall performance on the task precede the WSLS analysis.

We have moved the information regarding model fitting from Appendix 5 to the Materials and methods section of the main manuscript. We have also created a new subsection “Model validation” in which we report the similarity between the model simulations and participants’ responses. This section includes the figures that were previously in Appendix 5.